# Time intervals and routes to diagnosis for lung cancer in 10 jurisdictions: cross-sectional study findings from the International Cancer Benchmarking Partnership (ICBP)

Usha Menon ,[1] Peter Vedsted,[2] Alina Zalounina Falborg ,[2] Henry Jensen ,[2] Samantha Harrison,[3] Irene Reguilon ,[3] Andriana Barisic,[4] Rebecca J Bergin,[5,6] David H Brewster ,[7,8] John Butler,[9] Odd Terje Brustugun,[10] Oliver Bucher,[11] Victoria Cairnduff,[12] Anna Gavin,[12] Eva Grunfeld,[13] Elizabeth Harland,[11] Jatinderpal Kalsi,[1] Anne Kari Knudsen,[14,15] Mats Lambe,[16,17] Rebecca-Jane Law,[18] Yulan Lin,[14,15] Martin Malmberg,[19] Donna Turner,[20] Richard D Neal,[18,21] Victoria White,[5,22] David Weller[7]

UM, PV and DW contributed equally.

For numbered affiliations see end of article.

**Correspondence to**
Professor Usha Menon;
u.menon@ucl.ac.uk

## ABSTRACT

**Objective** Differences in time intervals to diagnosis and treatment between jurisdictions may contribute to previously reported differences in stage at diagnosis and survival. The International Cancer Benchmarking Partnership Module 4 reports the first international comparison of routes to diagnosis and time intervals from symptom onset until treatment start for patients with lung cancer.

**Design** Newly diagnosed patients with lung cancer, their primary care physicians (PCPs) and cancer treatment specialists (CTSs) were surveyed in Victoria (Australia), Manitoba and Ontario (Canada), Northern Ireland, England, Scotland and Wales (UK), Denmark, Norway and Sweden. Using Wales as the reference jurisdiction, the 50th, 75th and 90th percentiles for intervals were compared using quantile regression adjusted for age, gender and comorbidity.

**Participants** Consecutive newly diagnosed patients with lung cancer, aged ≥40 years, diagnosed between October 2012 and March 2015 were identified through cancer registries. Of 10 203 eligible symptomatic patients contacted, 2631 (27.5%) responded and 2143 (21.0%) were included in the analysis. Data were also available from 1211 (56.6%) of their PCPs and 643 (37.0%) of their CTS.

**Primary and secondary outcome measures** Interval lengths (days; primary), routes to diagnosis and symptoms (secondary).

**Results** With the exception of Denmark (−49 days), in all other jurisdictions, the median adjusted total interval from symptom onset to treatment, for respondents diagnosed in 2012–2015, was similar to that of Wales (116 days). Denmark had shorter median adjusted primary care interval (−11 days) than Wales (20 days); Sweden had shorter (−20) and Manitoba longer (+40) median adjusted diagnostic intervals compared with Wales (45 days). Denmark (−13), Manitoba (−11), England (−9)

## Strengths and limitations of this study

► This is the first study to use standardised survey methods and definitions to systematically examine key intervals from patients first noticing symptoms or bodily changes until the start of treatment for lung cancer across multiple jurisdictions.

► Recall bias was minimised by the triangulation of different data sources and by patients completing the questionnaire within a limited time window (median 5 months) after the cancer diagnosis.

► A key limitation, as with all questionnaire-based studies, was selection and non-response bias, which varied across jurisdictions.

► Recruitment of patients up to 9 rather than 6 months after diagnosis might have magnified selection bias due to high mortality in lung cancer but a sensitivity analysis suggests that this did not impact on the results.

► The comparisons for Norway and Victoria are limited by small sample size and inclusion of only surgical patients, respectively.

and Northern Ireland (−4) had shorter median adjusted treatment intervals than Wales (43 days). The differences were greater for the 10% of patients who waited the longest. Based on overall trends, jurisdictions could be grouped into those with trends of reduced, longer and similar intervals to Wales. The proportion of patients diagnosed following presentation to the PCP ranged from 35% to 75%.

**Conclusion** There are differences between jurisdictions in interval to treatment, which are magnified in patients with lung cancer who wait the longest. The data could help jurisdictions develop more focused lung cancer policy and

targeted clinical initiatives. Future analysis will explore if these differences in intervals impact on stage or survival.

## INTRODUCTION

Lung cancer is the most common cancer worldwide, with nearly 1.83 million cases diagnosed in 2012, and is the leading cause of cancer death globally, accounting for 19% of cancer deaths.[1] Survival is typically low, with 5-year survival in Europe, North America and Australia <20%.[2 3] A key factor is diagnosis at advanced stage. Reasons for this are multifaceted and include delays due to the atypical nature of some presenting symptoms, poor sensitivity of chest X-rays and physicians not acting quickly enough.[4] Within European countries, differences of 12 and 5 percentage points in 1-year and 5-year relative survival, respectively, have been reported for lung cancers diagnosed between 1999 and 2007.[5] This and other international comparisons raises the possibility of additional contributory factors such as variations in referral patterns, access to diagnostic tests and delays in treatment.[6]

One way of addressing this is to chart the patient journey from first noticing symptoms to treatment start. Many national studies using different methodologies have reported on time intervals to treatment of lung cancer, and there are reviews that have looked at international time frame comparisons.[7–24] However, as far as we are aware, there is no study that has undertaken international comparisons of timeliness across multiple countries using the same methodology.

The International Cancer Benchmarking Partnership (ICBP) was established to explore differences in cancer outcomes and their causes in countries with comparable wealth and universal access to healthcare.[25] We report results from ICBP Module 4 (ICBP M4) on differences in time intervals and routes to diagnosis in symptomatic patients with lung cancer from 10 jurisdictions in Australia, Canada, Denmark, Norway, Sweden and the UK.

## METHODS

Methods have been previously detailed.[26] In brief, in each of the 10 participating jurisdictions (Victoria (Australia), Manitoba, Ontario (Canada), Denmark, Norway, Sweden Northern Ireland, England, Scotland and Wales), consecutive patients aged ≥40 years, newly diagnosed with malignant lung or bronchus cancer (ICD-10 (10th revision of the International Statistical Classification of Diseases and Related Health Problems): C34.0-C34.9; ICD-O-3 behaviour code/3) were identified by the cancer registry using validated methods (hospital episode, cancer registration and pathology). Exclusion criteria included previous lung or synchronous cancers. Patients with a previous non-lung primary cancer were eligible. Target recruitment was 200 symptomatic patients per jurisdiction.

Following a vital status check, cancer registries posted the patient questionnaire (online supplementary appendix A1) either: (1) to the relevant primary care physician (PCP) who then forwarded the preaddressed envelope to the patient after confirmation that the person was aware of the diagnosis and not deemed too sick/anxious to participate in the survey (Wales, England and Scotland) or (2) to the patient directly or via the research team (remaining seven jurisdictions). In an attempt to decrease attrition and recall bias, the protocol initially specified that all patient questionnaires should be completed within 6 months of diagnosis. As there were administrative delays in cancer notification, this was extended to 9 months.

On receipt of a completed patient questionnaire, in all jurisdictions except Sweden, the relevant PCP and cancer treatment specialist (CTS) were sent questionnaires (online supplementary appendix A.2 and A.3). Specialists provided information on diagnosis and start date of treatment. The latter was collected directly from registry records in Northern Ireland and clinical databases in Denmark. Manitoba did not provide specialist data. Date of diagnosis and stage was also collected where possible through cancer registries. Information on the types of treatment (surgery, chemotherapy, radiotherapy and other) were obtained from the patient survey.

### Data handling

Data were recoded centrally to ensure that the same explicit rules were applied throughout. Patients in whom age, date of diagnosis or consent were missing were excluded from analyses. Rules were used to combine data from the different sources in a standardised way that ensured reproducibility and transparency (online supplementary appendix B). The rules employed a 'hierarchy' principle in terms of the order in which different data sources were used and included imputation rules based on the available data. The exact rule was guided by the measure in question, for example, patient interval was collected primarily from the patient questionnaire whereas primary care time-points were collected from the PCP questionnaire. We applied rules for outliers and implausible measures (eg, negative time intervals were recorded to zero-days and intervals longer than a year to 365 days).

### Routes to diagnosis and symptoms prompting physician visit

These were derived from patient and PCP responses. Symptoms were coded by two PCP authors (DW and PV) into 'lung cancer specific' or 'other' (online supplementary appendix C1).

### Time intervals

Time intervals were derived using the checklist for the Aarhus Statement.[27] The following time-points were used to calculate the corresponding time intervals (figure 1):
► First noticing symptoms.
► First presentation to healthcare

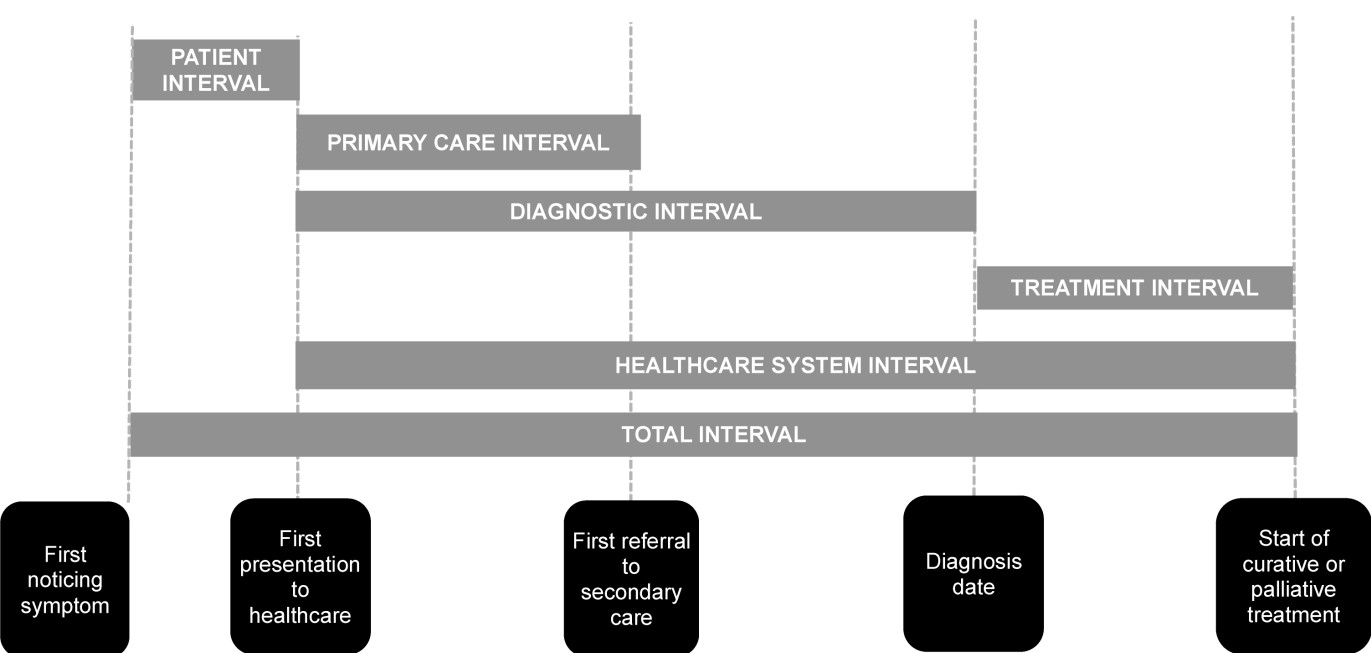

**Figure 1** Time intervals from onset of symptoms to start of treatment based on the Aarhus Statement.

► First referral to secondary care.
► Diagnosis date.
► Start of curative or palliative treatment.

All time-points were validated if there were obvious inconsistencies and negative intervals were set to 0 days. All intervals were truncated at 365 days. Missing data were imputed based on specific rules to ensure that the direction of a possible misclassification bias was known.

### Demographics
Health status was measured using the self-reported general health item from Short Form 36 health survey.[28] Comorbidity was defined as one of four patient or PCP-reported diseases (heart or lung disease, stroke or diabetes) and categorised into: 'none', 'medium' (one or two) or 'high' (three or four). Level of educational attainment was categorised as 'low' (vocational school or lower) and 'high' (university). Stage data (tumour, node and metastasis (TNM) classification) was grouped as I, II, III, IV or missing.[29]

### Statistical analysis
Patient characteristics across jurisdictions were compared using Kruskal-Wallis test for continuous and ordinal data. For nominal data, we used Pearson's $\chi^2$ test and Fisher's exact test (if more than 20% of expected cell counts were less than five or at least one expected cell count was 0). The differences in intervals between the jurisdictions were estimated using quantile regression, as this method allows for a comparison across the whole distribution of length of the interval.[30] As we were interested in a measure of central tendency of length of the interval and in long and very long intervals, the focus of the study was on the 50th (median), 75th and 90th interval percentiles. Wales was chosen as the reference jurisdiction as it had the lowest lung cancer survival in ICBP Module 1 analysis.[10] Since

the length of the interval in days is a continuous measure that has been rounded, we applied the quantile regression analysis on the smoothed quantiles; the method based on the smoothed quantiles is recommended for analyses of discrete (count) data.[31] In STATA, this method is implemented in the 'qcount' procedure.[32] Parameters were calculated with 1000 jittered samples. For all interval analyses, the differences in intervals were calculated as marginal effects after quantile regression by setting the continuous covariate (age) to their mean values and the categorical covariates (sex and comorbidity) to their modes. Significance level was set to ≤0.05 with 95% CIs calculated where appropriate. Statistical analyses were carried out using STATA V.14.

### Sensitivity and validity analyses
All analyses were repeated using only: (1) those who fulfilled the 6-month cut-off criteria for interval from diagnosis to questionnaire completion and (2) patient data. The effect of excluding patients for whom at least one interval was missing was investigated. We also repeated the analysis after omitting time intervals that were negative or over 365 days. Agreement between the different data sources (registry, patient, PCP (except in Sweden) and CTS (except in Sweden and Manitoba)) was measured by Lin's concordance correlation coefficient (CCC).[33]

### Patient involvement
The research questions for this survey drew on an extensive literature relating to diagnosis and treatment delays leading to negative patient experiences. While patient experience was not a primary outcome measure for this study, patients were given the opportunity to comment on their experience through questionnaire free-text response options (under separate analysis). Patients were

involved in the piloting of study instruments to ascertain if recruitment and questionnaire content and dissemination strategies were appropriate, as described elsewhere.[26] Each jurisdiction has committed to communicating the findings and local implications of this study to organisations representing their study participants.

## RESULTS

Of 14 583 patients with lung cancer, diagnosed between October 2012 and March 2015 who were alive when identified, 70% (10 203/14 583) were contacted (table 1). Of 4380 not contacted, 3367 (77%) were from England, Wales and Scotland. Major reasons reported by the PCP for not forwarding the survey included patients being terminally ill, not aware of cancer diagnosis at the time of request, having cognitive or visual impairment, language/communication difficulties, no longer at the address, not wishing to take part in research and a small number not having the index cancer. In addition, patents identified were not contacted in England as the target recruitment had been exceeded. For the non-UK jurisdictions, the main reasons for not contacting patients were the patient having died or no longer at the address.

A total of 2631 (27.5% of contacted, 18% of eligible) completed the patient questionnaire at a median of 5 months (range: 0.1–9) after diagnosis (table 2). The response rate of contacted patients varied from 11.1% (146/1318) in Norway to 61.8% (333/539) in Denmark. Responding patients were more likely to be aged 60–79 years with less advanced stage (table 1) and alive 1 year postdiagnosis (data not shown). Of the 2631 responses, 2143 (81.5%) were included in the analyses which equates to 14.7% (2143/14 583) of eligible patients. The key reason respondents were excluded were local oversampling (43.9%; 214/488) for additional analyses (table 1). In Victoria, the registry was only able to contact patients who had undergone surgery while the sample size in Norway was limited (n=88) due to delays in securing appropriate approvals.

### Baseline characteristics

Patient characteristics are detailed in table 2. The cohort was predominantly white (95%), median age 70 years (IQR 64–75) with 82% reporting 'low' levels of education. Norway provided stage data classified as local, regional and distant, which could not be converted to TNM stage data.

Ontario was the only jurisdiction with more female (65%) than male respondents. While self-reported health differed significantly, with Welsh patients (9%) reporting twice as high 'poor health' rates than English or Swedish (4%) and eightfold that of Manitoba (2%), there was no difference in self-reported comorbidity rates. Even after the exclusion of Victoria, there was significant variation in early stage (I/II) disease, which ranged from 25% (Wales) to 59% (Ontario), and surgical resection rates, which varied from 27% (Wales) to 58% (Ontario) (table 2).

### Routes to diagnosis

Results are detailed in table 3. Over half (55%) were diagnosed following presentation to the PCP of whom 63% (range: 29% Norway–82% Wales; data not shown) were urgently referred with a suspicion of cancer based on the PCP questionnaire.

### Symptoms prompting visit to physician

The median number of patient-reported symptoms were 2 (IQR 1–3). Across jurisdictions, the most common patient-reported symptoms were persistent cough (39%), breathlessness (37%) and fatigue (27%), although there was significant variation in proportion of patients presenting with individual symptoms (table 4).

The PCPs reported a median of 1 (IQR 1–2) symptom at first presentation, with the most common being persistent cough (39%). Across jurisdictions, the reporting of other symptoms by the PCP was significantly lower compared with patients, especially fatigue (4%) and weight loss (8%). When the analysis was restricted to the cohort where both patient and PCP had completed the survey, this difference persisted. Unlike patients, there was minimal variation in PCP reporting of symptoms, with significant differences limited to 'no symptoms', 'other symptoms not previously listed' and weight loss. Overall, 64% of symptoms were labelled as 'cancer specific' by the study PCPs (table 4).

### Time intervals

The observed time intervals are shown in online supplementary appendix C2 and are summarised in figure 2. Based on overall trends, jurisdictions could be grouped into those with reduced, longer and similar intervals to Wales (table 5). It was not possible to interpret variations observed in Norway (small sample size) and Victoria (surgical cohort). In the remaining jurisdictions, there was no difference in the median adjusted patient interval compared with Wales. Denmark had shorter median adjusted primary care interval (−11 days); Sweden had shorter (−20) and Manitoba had longer (+40) diagnostic intervals compared with Wales. Denmark (−13), Manitoba (−11), England (−9) and Northern Ireland (−4) had shorter treatment intervals. The median adjusted total interval was shorter only in Denmark (table 5, figure 2). The differences were greater for the 90th percentile. The total interval in patients who waited longest (90th percentile) was significantly shorter in two jurisdictions (Denmark: −142 days; England: −28 days) compared with Wales.

### Sensitivity and validity analyses

The estimates of routes to diagnosis, time intervals and regression analysis trends were not significantly altered by changing the cut-off to 6 months or using only patient data (results not shown).

Online supplementary appendix C3 details which sources were used based on the standardisation rules to define dates and also how often a day in the date was

**Table 1** Overview of the flow of patients from eligibility to analyses for all jurisdictions

| Jurisdiction | Patients approached via PCP | | | Patient approached directly by registries/research teams | | | | | | | Total |
| | Wales | England | Scotland | Northern Ireland | Denmark | Manitoba | Ontario | Sweden | Norway | Victoria | |
| | n (%) | n (%) | n (%) | n (%) | n (%) | n (%) | n (%) | n (%) | n (%) | n (%) | n (%) |
| Eligible patients*† | 1811 (100) | 2517 (100) | 1366 (100) | 620 (100) | 539 (100) | 980 (100) | 4080 (100) | 493 (100) | 1318 (100) | 859 (100) | 14583 (100) |
| Packs sent to PCP‡§ | 1811 (99.7) | 1759 (69.9) | 1137 (83.2) | | | | | | | | 4707 (82.7) |
| Pack not forwarded by PCP | 547 (30.1) | 255 (14.5) | 201 (17.7) | | | | | | | | 1003 (21.3) |
| Unsure if pack forwarded by PCP | 531 (29.2) | 559 (31.8) | 234 (20.6) | | | | | | | | 1324 (28.1) |
| Patients contacted by PCP‡§ | 733 (40.4) | 945 (53.7) | 702 (61.7) | | | | | | | | 2380 (50.6) |
| Patients approached directly¶ | | | | 614 (99.0) | 539 (100) | 745 (76.0) | 3687 (90.4) | 493 (100) | 1200 (91) | 545 (63.4) | 7823 (88) |
| Patient died | | | | 6 (1.0) | 0 (0.0) | 103 (13.8) | 249 (6.8) | 0 (0.0) | 0 (0.0) | 0 (0.0) | 358 (4.6) |
| No address | | | | 0 (0.0) | 0 (0.0) | 9 (1.2) | 255 (6.9) | 0 (0.0) | 0 (0.0) | 0 (0.0) | 264 (3.4) |
| Other | | | | 0 (0.0) | 0 (0.0) | 6 (0.8) | 215 (5.8) | 0 (0.0) | 0 (0.0) | 0 (0.0) | 221 (2.8) |
| Patient responses (% of eligible patients)¶ | 223 (12.3) | 261 (10.4) | 235 (17.2) | 226 (36.5) | 333 (61.8) | 205 (20.9) | 572 (14.0) | 217 (44) | 146 (11.1) | 213 (24.8) | 2631 (18) |
| Patient responses (% of contacted)¶ | 223 (30.4) | 261 (27.6) | 235 (33.5) | 226 (37.2) | 333 (61.8) | 205 (32.7) | 572 (19.3) | 217 (44) | 146 (12.2) | 213 (39.1) | 2631 (27.5) |
| Extra sample for local purpose | 0 (0.0) | 0 (0.0) | 0 (0.0) | 0 (0.0) | 0 (0.0) | 0 (0.0) | 214 (37.4) | 0 (0.0) | 0 (0.0) | 0 (0.0) | 214 (8.1) |
| Other | 0 (0.0) | 0 (0.0) | 35 (14.9) | 25 (11.1) | 38 (11.4) | 0 (0.0) | 43 (7.5) | 0 (0.0) | 0 (0.0) | 3 (1.4) | 144 (5.5) |

Continued

**Table 1** Continued

| Jurisdiction | Patients approached via PCP | | | | | | | Patient approached directly by registries/research teams | | | Total |
|---|---|---|---|---|---|---|---|---|---|---|---|
| | Wales | England | Scotland | Northern Ireland | Denmark | Manitoba | Ontario | Sweden | Norway | Victoria | |
| | n (%) | n (%) | n (%) | n (%) | n (%) | n (%) | n (%) | n (%) | n (%) | n (%) | n (%) |
| Patient surveys submitted for analyses** ‡ | 223 (100) | 261 (100) | 200 (85.1) | 201 (88.9) | 295 (88.6) | 205 (100) | 315 (55.1) | 217 (100) | 146 (100) | 210 (98.6) | 2273 (86.4) |
| Excluded for analyses – total | 12 (5.4) | 9 (3.4) | 2 (1.0) | 1 (0.5) | 10 (3.4) | 3 (1.5) | 27 (8.6) | 6 (2.8) | 58 (39.7) | 2 (1.0) | 130 (5.7) |
| Previous cancer | 0 (0.0) | 5 (1.9) | 0 (0.0) | 0 (0.0) | 0 (0.0) | 0 (0.0) | 3 (1.0) | 0 (0.0) | 0 (0.0) | 0 (0.0) | 8 (0.4) |
| Unknown date of consent or diagnosis | 0 (0.0) | 0 (0.0) | 0 (0.0) | 0 (0.0) | 5 (1.7) | 0 (0.0) | 1 (0.3) | 6 (2.8) | 4 (2.7) | 0 (0.0) | 16 (0.7) |
| Consent too late/too early | 12 (5.4) | 4 (1.5) | 2 (1.0) | 1 (0.5) | 5 (1.7) | 3 (1.5) | 22 (7.0) | 0 (0.0) | 33 (22.6) | 2 (1.0) | 84 (3.7) |
| Other | 0 (0.0) | 0 (0.0) | 0 (0.0) | 0 (0.0) | 0 (0.0) | 0 (0.0) | 1 (0.3) | 0 (0.0) | 21 (14.4) | 0 (0.0) | 22 (1.0) |
| Patients included in analyses†† (% of forwarded surveys) | 211 (94.6) | 252 (96.6) | 198 (99.0) | 200 (99.5) | 285 (96.6) | 202 (98.5) | 288 (91.4) | 211 (97.2) | 88 (60.3) | 208 (99.0) | 2143 (94.3)‡‡ |
| PCP surveys§§ (% of analysed patients) | 133 (63.0) | 196 (77.8) | 149 (75.3) | 181 (90.5) | 218 (76.5) | 109 (54.0) | 93 (32.5) | n/a‡ 27 | (30.7) | 105 (50.5) | 1211 (56.6)†† |
| Specialist surveys¶¶ (% of analysed patients) | 98 (46.4) | 153 (60.7) | 106 (53.5) | n/a*** 149 | (52.3) | n/a‡ 62 | (21.7) | n/a‡ 20 | (22.7) | 55 (26.4) | 643 (37.0) |

Continued

**Table 1** Continued

| Jurisdiction | Patients approached via PCP | | | | Patient approached directly by registries/research teams | | | | | | |
| --- | --- | --- | --- | --- | --- | --- | --- | --- | --- | --- | --- |
| | Wales | England | Scotland | Northern Ireland | Denmark | Manitoba | Ontario | Sweden | Norway | Victoria | Total |
| | n (%) | n (%) | n (%) | n (%) | n (%) | n (%) | n (%) | n (%) | n (%) | n (%) | n (%) |

*Eligible as per protocol: individual aged 40 years or more, with cancer of the lung or bronchus (ICD-10 code: C34.0-C34.9; behaviour code ICD-O 3) but no synchronous primary cancer or priorhistory of lung cancer, alive at identification who completed consent to participate within 9 months of diagnosis.
†In some jurisdictions, some 'eligible' patients had preopted out from being contacted and a small number where PCP information was not available.
‡Percentages of eligible patients.
§Maximum of potentially contacted patients. That is, sum of packs forwarded by PCP and packs unsure if forwarded by PCP.
¶Percentages of patients contacted by PCP (see note d) for Wales, England and Scotland or percentages of patients contacted directly by a registry, excluding non-accessible patients due to death or no patient addresses (all other jurisdictions).
**Percentages of patient responses.
††Denominator=total number of forwarded cases excluding patients not included in analytic sample in Ontario.
‡‡Data not collected in this jurisdiction.
§§Denominator r=total number of analysed cases excluding patientsfrom Sweden.
¶¶Denominator r=total number of analysed cases excluding patients from Sweden, Manitoba and Northern Ireland.
***Data obtained from registries instead in Northern Ireland and Denmark.

imputed. With regards to the dates of first presentation to healthcare (CCC=0.91), diagnosis (CCC ≥0.93) and treatment (CCC=0.94), there was adequate agreement between all data sources where the data on these dates was collected. Agreement between patient versus PCP for dates of first presentation to healthcare (CCC=0.91) and diagnosis (CCC=0.93) was also adequate as was agreement between patient versus CTS for dates of diagnosis (CCC=0.94) and treatment (CCC=0.94).

Omitting time intervals that were negative or over 365 days (online supplementary appendix C4) led to change in direction of difference, which was non-significant in long intervals (75th or 90th percentile) between Wales and jurisdictions in four cases: Norway and Victoria (patient interval), Northern Ireland (diagnostic interval) and England (total interval). All other results were similar to the main results.

## DISCUSSION
### Main findings
This is the first international study we are aware of comparing lung cancer routes and time intervals. With the exception of Denmark, in all other jurisdictions, the median total interval from symptom onset to treatment, for respondents diagnosed in 2012–2015, was similar to that of Wales, which is the reference. However, there were jurisdiction specific differences in patient, diagnostic and treatment intervals, especially for the 10% of patients who waited the longest. Based on overall trends, jurisdictions could be grouped into those with trends of reduced, longer and similar intervals to Wales.

Across jurisdictions, all symptoms other than persistent cough were less frequently reported by the PCP when compared with patients. This was especially true for fatigue and weight loss. One in four patients reported incidental diagnosis and 1 in 10 were diagnosed following a visit to the emergency (Accident and Emergency (A&E)) department.

### Strengths and weaknesses
Our study helps address the shortcomings of current international comparisons across multiple national studies with significant variation in methodology including differences in definition of intervals. Strengths of our study include: (1) use of the same methodology across countries; (2) use of cancer registries to identify consecutive newly diagnosed patients; (3) use of standardised questionnaires; (4) inclusion of PCP and CTS questionnaires enriched by registry data; (5) minimal data interpretation by the local teams with all data cleaning performed in a standardised manner centrally; and (6) triangulation with comprehensive data rules to ensure validity, consistency and preserve statistical precision.[21] Recall bias was minimised by the triangulation of different data sources and by patients completing the questionnaire within a limited time window (median 5 months) after the cancer diagnosis.

**Table 2** The characteristics of eligible patients by jurisdictions and overall

| | Wales | England | Scotland | Northern Ireland | Denmark | Manitoba | Ontario | Sweden | Norway | Victoria | Overall | P value* |
|---|---|---|---|---|---|---|---|---|---|---|---|---|
| | (n=211) | (n=252) | (n=198) | (n=200) | (n=285) | (n=202) | (n=288) | (n=211) | (n=88) | (n=208) | (n=2143) | |
| Date of diagnosis of first patient | 3. april 2013 | 28. january 2013 | 26. april 2013 | 22. januar y2013 | 16. may 2013 | 10. october 2012 | 8. october 2013 | 1. october 2013 | 16. january 2014 | 8. january 2013 | 10. october 2012 | |
| Date of diagnosis of last patient | 25. september 2014 | 14. august 2013 | 4. december 2013 | 2. december 2014 | 13. november 2013 | 27. march 2015 | 1. october 2014 | 30. may 2014 | 21. january 2015 | 28. december 2014 | 27. march 2015 | |
| Interval from diagnosis date of first patient to last patient in months (recruitment period) | 18 | 7 | 7 | 23 | 6 | 30 | 12 | 8 | 12 | 24 | 30 | |
| Median (range) interval diagnosis to questionnaire completion in months | 5 (0.6,9) | 5 (3,9) | 5 (1,9) | 4 (0.1,9) | 5 (2,8) | 6 (4,9) | 6 (4,9) | 4 (3,8) | 7 (5,9) | 5 (0.2,8) | 5 (0.1,9) | |
| **Age years** | | | | | | | | | | | | |
| Median (IQR) | 71 (65–77) | 71 (65–77) | 70 (64–76) | 69 (62–75) | 70 (63–74) | 70 (63–77) | 70 (64–75) | 70 (63–75) | 69 (64–73) | 68 (63–73) | 70 (64–75) | 0.010† |
| **Sex** n(%) | | | | | | | | | | | | |
| Male | 127(60) | 134(53) | 100(51) | 105(53) | 151(53) | 100(50) | 131(45) | 111(53) | 46(52) | 112(54) | 1117(52) | 0.159‡ |
| **Health State** n(%) | | | | | | | | | | | | |
| Good | 135(64) | 176(70) | 131(66) | 126(63) | 190(67) | 156(77) | 220(76) | 152(72) | 50(57) | 163(78) | 1499(70) | <0.001§† |
| Fair | 55(26) | 59(23) | 52(26) | 51(26) | 62(22) | 41(20) | 47(16) | 49(23) | 32(36) | 33(16) | 481(22) | <0.001¶‡ |
| Poor | 20(9) | 11(4) | 14(7) | 16(8) | 18(6) | 4(2) | 16(6) | 9(4) | 6(7) | 10(5) | 124(6) | |
| Missing | 1(0.5) | 6(2) | 1(0,5) | 7(4) | 15(5) | 1(0.5) | 5(2) | 1(0.5) | 0(0) | 2(1) | 39(2) | |
| **Comorbidity**\**n(%) | | | | | | | | | | | | |
| No | 92(44) | 113(45) | 98(49) | 82(41) | 111(39) | 101(50) | 136(47) | 114(54) | 35(40) | 98(47) | 980(46) | 0.029§† |
| Medium | 111(53) | 129(51) | 92(46) | 103(52) | 157(55) | 93(46) | 132(46) | 87(41) | 48(55) | 100(48) | 1052(49) | 0.032¶‡ |
| High | 7 (3) | 9 (4) | 8 (4) | 15(8) | 11 (4) | 6 (3) | 18(6) | 6 (3) | 5 (6) | 8 (4) | 93(4) | |
| Missing | 1 (0.5) | 1 (0.4) | 0(0) | 0(0) | 6 (2) | 2 (1) | 2 (0.7) | 4 (2) | 0(0) | 2 (1) | 18 (0.8) | |
| **Education** n(%) | | | | | | | | | | | | |
| Low | 172(82) | 217(86) | 174(88) | 166(83) | 232(81) | 170(84) | 224(78) | 159(75) | 65(74) | 181(87) | 1760(82) | <0.001§† |
| High | 11(5) | 15(6) | 11(6) | 17(9) | 15(5) | 21(10) | 55(19) | 48(23) | 14(16) | 24(12) | 231(11) | <0.001¶‡ |
| Missing | 28(13) | 20(8) | 13(7) | 17(9) | 38(13) | 11(5) | 9(3) | 4(2) | 9(10) | 3(1) | 152(7) | |
| **Ethnicity** n(%) | | | | | | | | | | | | |
| White | 205(97) | 249(99) | 197(99) | 196(98) | 267(94) | 173(86) | 261(91) | n/a | 87(99) | 199(96) | 1834(95) | <0.001§‡ |
| Asian | 1 (0.5) | 0 (0) | 1 (0.5) | 0 (0) | 0 (0) | 12(6) | 21(7) | n/a | 1 (1) | 6 (3) | 41(2) | <0.001¶‡ |

Continued

**Table 2** Continued

| | | | | | | | | | | | Total | p value | p value |
|---|---|---|---|---|---|---|---|---|---|---|---|---|---|
| Black | 0(0) | 1(0.4) | 0(0) | 0(0) | 1(0.4) | 0(0) | 1(0.3) | n/a | 0(0) | 0(0) | 3(0.2) | | |
| Other | 0(0) | 0(0) | 0(0) | 0(0) | 15(7) | 0(0) | 2(0.7) | n/a | 0(0) | 0(0) | 17(0.9) | | |
| Missing | 5(2) | 2(0.8) | 0(0) | 4(2) | 17(6) | 2(1) | 3(1) | n/a | 0(0) | 3(1) | 36(2) | | |
| **Smoking n(%)** | | | | | | | | | | | | | |
| Never | 13(6) | 18(7) | 12(6) | 19(10) | 15(5) | 22(11) | 31(11) | 41(19) | 11(13) | 33(16) | 215(10) | <0.001§‡ | <0.001¶‡ |
| Currently | 19(9) | 288(11) | 26(13) | 41(21) | 58(20) | 24(12) | 29(10) | 29(14) | 11(13) | 9(4) | 274(13) | | |
| In the past | 174(82) | 204(81) | 160(81) | 137(69) | 205(72) | 156(77) | 225(78) | 139(66) | 66(75) | 165(79) | 1631(76) | | |
| Missing | 5(2) | 2(0.8) | 0(0) | 3(2) | 7(2) | 0(0) | 3(1) | 2(0.9) | 0(0) | 1(0.5) | 23(1) | | |
| **Tumour stage – TNM n(%)** | | | | | | | | | | | | | |
| I | 26(12) | 68(27) | 49(25) | 42(21) | 74(26) | 83(41) | 133(46) | 59(28) | 3(3) | 124(60) | 661(31) | <0.001§† | <0.001¶‡ |
| II | 27(13) | 36(14) | 32(16) | 33(17) | 26(9) | 19(9) | 38(13) | 11(5) | 5(6) | 47(23) | 274(13) | | |
| III | 64(30) | 57(23) | 56(28) | 59(30) | 84(29) | 48(24) | 54(19) | 40(19) | 5(6) | 19(9) | 486(23) | | |
| IV | 62(29) | 80(32) | 50(25) | 62(31) | 94(33) | 48(24) | 54(19) | 94(45) | 3(3) | 13(6) | 560(26) | | |
| Missing | 32(15) | 11(4) | 11(6) | 4(2) | 7(2) | 4(2) | 9(3) | 7(3) | 72(82)†† | 5(2) | 162(8) | | |
| **Tumour stage – TNM n(%)** | | | | | | | | | | | | | |
| I/II | 53(25) | 104(41) | 81(41) | 75(38) | 100(35) | 102(51) | 171(59) | 70(33) | 8(9) | 171(82) | 764(39) | <0.001§‡ | <0.001¶‡ |
| III/IV | 126(60) | 137(54) | 106(54) | 121(61) | 178(62) | 96(48) | 108(38) | 134(64) | 8(9) | 32(15) | 1014(52) | | |
| Missing | 32(15) | 11(4) | 11(6) | 4(2) | 7(2) | 4(2) | 9(3) | 7(3) | 72(82)†† | 5(2) | 157(8) | | |
| **Treatment Surgery n(%)** | | | | | | | | | | | | | |
| Yes | 57(27) | 107(42) | 84(42) | 65(33) | 113(56) | 81(28) | 168(58) | 65(31) | 36(41) | 199(96) | 975(45) | <0.001§‡ | <0.001¶‡ |
| No | 61(29) | 56(22) | 55(28) | 94(47) | 44(22) | 90(32) | 111(39) | 79(37) | 28(32) | 5(2) | 623(29) | | |
| Missing | 93(44) | 89(35) | 59(30) | 41(21) | 45(22) | 114(40) | | 67(32) | 24(27) | 4(2) | 545(25) | | |
| **Treatment Chemo n(%)** | | | | | | | | | | | | | |
| Yes | 105(50) | 125(50) | 98(49) | 93(47) | 93(46) | 107(37) | 133(63) | 50(57) | 63(30) | 1026(48) | | <0.001§‡ | <0.001¶‡ |
| No | 41(19) | 51(20) | 46(23) | 65(33) | 62(31) | 172(60) | 37(18) | 18(20) | 137(66) | 674(32) | | | |
| Missing | 65(31) | 76(30) | 54(27) | 42(21) | 47(23) | 81(28) | 9(3) | 41(19) | 20(23) | 8(4) | 443(21) | | |
| **Treatment Radio n(%)** | | | | | | | | | | | | | |
| Yes | 82(39) | 68(27) | 76(38) | 72(36) | 81(40) | 114(40) | 98(34) | 41(47) | 29(14) | 731(34) | | <0.001§‡ | <0.001¶‡ |
| No | 50(24) | 72(29) | 50(25) | 77(39) | 71(35) | 69(24) | 180(63) | 22(25) | 155(75) | 815(38) | | | |
| Missing | 79(37) | 112(44) | 72(36) | 51(26) | 50(25) | 102(36) | 10(3) | 25(28) | 24(12) | 597(28) | | | |

Continued

**Table 2** Continued

| Treatment Other n(%) | | | | | | | | | | | | |
|---|---|---|---|---|---|---|---|---|---|---|---|---|
| Yes | 10(5) | 14(6) | 14(7) | 11(6) | 30(11) | 18(9) | 9(3) | 0(0) | 7(8) | 16(8) | 129(6) | <0.001§‡ ‡‡ |
| No | 48(23) | 63(25) | 45(23) | 69(35) | 255(89) | 3(1) | 261(91) | 0(0) | 2(2) | 138(66) | 884(41) | <0.001¶‡ ‡‡ |
| Missing | 153(72) | 175(69) | 139(70) | 120(60) | 0(0) | 181(90) | 18(6) | 211(100) | 79(90) | 54(26) | 1130(53) | |

*Excluding Norway.
†Differences between jurisdictions were tested by the Kruskal-Wallis test.
‡Differences between jurisdictions were tested by the Pearson's $\chi^2$ test.
§Missing category is excluded.
¶Missing category is included.
**Comorbidity coded as none=no reported, medium=1-2 reported and high=3+ reported.
††This included cases which could not be mapped as they were classified as per Cancer Registry of Norway into local (stage I), regional (stage II-III) and distant (stage IV).
‡‡Excluding Victoria.
IQR, inter-quartile range; TNM, tumour, node and metastasis.

A key limitation, as with all questionnaire-based studies, was both selection and non-response bias that varied across jurisdictions and has implications for interpretation and generalisation of findings.[34] In comparing intervals, we adjusted for age, sex and comorbidity but were unable to adjust for ethnicity and education due to different classification systems. Recall bias was minimised by the triangulation of different data sources and by patients completing the questionnaire within a limited time window (median 5 months) after the cancer diagnosis. Recruitment of patients up to 9 rather than 6 months after diagnosis might have magnified the selection bias due to high mortality.[35] However, sensitivity analysis suggests that this did not impact on the results. Categorising presenting symptoms into indicative or not was done pragmatically as existing guidelines for lung cancer investigation vary across ICBP jurisdictions.[36] In Norway and Victoria, a small sample size and restriction of eligibility to only surgical patients, respectively, made comparison difficult. Nonetheless, significant differences in these two jurisdictions compared with Wales were largely limited to the treatment interval alone.

There was variation in stage distribution across jurisdictions. While this may be partly related to the varying response rate, true differences in lung cancer stage have been noted on analysis of registry data of patients diagnosed between 2004 and 2007.[6] The high lung cancer mortality and self-selection are likely to have contributed to an over-representation of early stage disease and tumours treated with surgical resection. This suggests that true variation may well be higher than that reported in this cohort of 'healthier early stage' patients.

### Comparison with other studies

The most common patient-reported symptoms, in keeping with the literature, were persistent cough, breathlessness, fatigue and weight loss, with one in five reporting 'no symptoms'.[18] Only a minority (11%) of our respondents reported coughing blood or bloody sputum/spit, which is the only consistent predictor of lung cancer.[37] While haemoptysis was reported in a prospective survey (England 2011–2012) by 22% of patients with lung cancer identified through respiratory clinics, it was a presenting symptom in only 5% of cases.[11]

The median number of symptoms reported by patients was more than that reported by the PCP in all jurisdictions. This was especially so for fatigue and weight loss. A number of factors could have contributed to this: patients not listing all symptoms at presentation, patients having a different understanding/recall of their symptoms post diagnosis and PCPs only recording key symptoms such as cough. Further research on under-reporting of systemic symptoms such as fatigue and weight loss is warranted.

As lung cancer mortality is higher in patients attending emergency (A&E) departments, the rates are often compared in an attempt to understand international

**Table 3** Routes to diagnosis of patients with lung cancer for each jurisdiction

| | Wales (N=211) | England (N=252) | Scotland (N=198) | Northern Ireland (N=200) | Denmark (N=285) | Manitoba (N=202) | Ontario (N=288) | Sweden (N=211) | Norway (N=88) | Victoria (N=208) | Total (N=2143) |
|---|---|---|---|---|---|---|---|---|---|---|---|
| Symptoms prompting visit to PCP | 109 (52) | 150 (60) | 128 (65) | 131 (66) | 170 (60) | 101 (50) | 91 (32) | 65 (31) | 35 (40) | 106 (51) | 1086 (51) |
| Symptoms prompting emergency (A&E) department visit* | 11 (5) | 18 (7) | 12 (6) | 22 (11) | 21 (7) | 25 (12) | 39 (14) | 18 (9) | 3 (3) | 6 (3) | 175 (8) |
| Symptoms prompting visit to PCP and emergency (A&E) department* | 10 (5) | 4 (2) | 7 (4) | 18 (9) | 8 (3) | 15 (7) | 12 (4) | 8 (4) | 5 (6) | 3 (1) | 90 (4) |
| Incidental diagnosis in course of investigation/ treatment for another problem† | 68 (32) | 37 (15) | 36 (18) | 19 (10) | 55 (19) | 57 (28) | 116 (40) | 107 (51) | 32 (36) | 90 (43) | 617 (29) |
| Unknown routes to diagnosis‡ | 10 (5) | 17 (7) | 11 (6) | 5 (3) | 16 (6) | 4 (2) | 14 (5) | 11 (5) | 8 (9) | 0 (0) | 96 (5) |
| Other§ | 3 (1) | 25 (10) | 3 (2) | 4 (2) | 14 (5) | 0 (0) | 16 (6) | 2 (1) | 5 (6) | 3 (1) | 75 (3) |
| Missing | 0 (0) | 1 (0.4) | 1 (0.5) | 1 (0.5) | 1 (0.4) | 0 (0) | 0 (0) | 0 (0) | 0 (0) | 0 (0) | 4 (0.2) |

All figures are n (%) unless otherwise stated.
*Emergency (A&E) route was ascribed when the patient-reported pathway to cancer diagnosis involving going or being taken to the emergency department or the PCP reported that the patient presented to emergency department with or without their involvement.
†This could be by PCP, another doctor or via hospital.
‡Includes cases where PCP or patient-reported routes to diagnosis as 'Other' or 'Missing' but also reported symptoms or duration of symptoms, or date of first symptom, or waiting time for PCP appointment.
§Includes cases where PCP or patient-reported routes to diagnosis as 'Other' and has not reported any symptoms or duration of symptoms, or date of first symptom, or waiting time for PCP appointment.
A&E, Accident and Emergency; PCP, primary care physician.

**Table 4** Symptoms experienced by patients and presenting symptoms noted by PCP for eligible patients

| Symptoms (reported by patient) | Wales (N=211) | England (N=252) | Scotland (N=198) | Northern Ireland (N=200) | Denmark (N=285) | Manitoba (N=202) | Ontario (N=288) | Sweden (N=211) | Norway (N=88) | Victoria (N=208) | Overall (N=2143) | P value* |
|---|---|---|---|---|---|---|---|---|---|---|---|---|
| Persistent cough | 113 (54) | 123 (49) | 97 (49) | 83 (42) | 97 (34) | 71 (35) | 125 (43) | 84 (40) | 10 (11) | 39 (19) | 842 (39) | <0.001 |
| Breathlessness | 109 (52) | 126 (50) | 82 (41) | 73 (37) | 99 (35) | 63 (31) | 119 (41) | 77 (36) | 12 (14) | 2 (12) | 784 (37) | <0.001 |
| Fatigue | 75 (36) | 79 (31) | 64 (32) | 61 (31) | 38 (13) | 55 (27) | 92 (32) | 60 (28) | 17 (19) | 42 (20) | 583 (27) | <0.001 |
| Weight loss | 38 (18) | 39 (15) | 44 (22) | 37 (19) | 41 (14) | 29 (14) | 36 (13) | 34 (16) | 9 (10) | 20 (10) | 327 (15) | 0.147 |
| Felt sick/vomiting/nausea/loss of appetite | 42 (20) | 33 (13) | 31 (16) | 24 (12) | 33 (12) | 22 (11) | 45 (16) | 28 (13) | 3 (3) | 20 (10) | 281 (13) | 0.132 |
| Coughing up blood-stained phlegm (sputum) | 35 (17) | 32 (13) | 24 (12) | 31 (16) | 21 (7) | 16 (8) | 27 (9) | 25 (12) | 5 (6) | 23 (11) | 239 (11) | 0.014 |
| Chest or shoulder pain | 23 (11) | 9 (4) | 18 (9) | 24 (12) | 10 (4) | 28 (14) | 43 (15) | 25 (12) | 4 (5) | 22 (11) | 206 (10) | <0.001 |
| Other symptoms not listed above | 51 (24) | 80 (32) | 59 (30) | 54 (27) | 63 (22) | 40 (20) | 30 (10) | 75 (36) | 26 (30) | 42 (20) | 520 (24) | <0.001 |
| No symptoms | 29 (14) | 24 (10) | 32 (16) | 21 (11) | 63 (22) | 50 (25) | 67 (23) | 36 (17) | 33 (38) | 75 (36) | 430 (20) | <0.001 |
| Missing | 1 (0.5) | 16 (6) | 6 (3) | 17 (9) | 31 (11) | 9 (4) | 4 (1) | 4 (2) | 12 (14) | 1 (0.5) | 101 (5) | <0.001 |
| **Presenting symptom (reported by PCP)** | (N=85) | (N=133) | (N=115) | (N=0) | (N=151) | (N=86) | (N=61) | (N=0) | (N=17) | (N=81) | (N=729) | |
| Persistent cough | 33 (39) | 57 (43) | 45 (39) | n/a | 67 (44) | 24 (28) | 18 (30) | n/a | 7 (41) | 33 (41) | 284 (39) | 0.093 |
| Breathlessness | 17 (20) | 27 (20) | 20 (17) | n/a | 27 (18) | 11 (13) | 11 (18) | n/a | 4 (24) | 13 (16) | 130 (18) | 0.803 |
| Fatigue | 1 (1) | 5 (4) | 3 (3) | n/a | 9 (6) | 2 (2) | 2 (3) | n/a | 1 (6) | 3 (4) | 26 (4) | 0.539† |
| Weight loss | 5 (6) | 9 (7) | 15 (13) | n/a | 19 (13) | 3 (3) | 2 (3) | n/a | 0 (0) | 5 (6) | 58 (8) | 0.027 |
| Felt sick/vomiting/nausea/loss of appetite | 1 (1) | 1 (0.8) | 2 (2) | n/a | 6 (4) | 2 (2) | 0 (0) | n/a | 0 (0) | 0 (0) | 12 (2) | 0.418† |
| Coughing up blood-stained phlegm (sputum) | 8 (9) | 10 (8) | 10 (9) | n/a | 7 (5) | 2 (2) | 3 (5) | n/a | 0 (0) | 7 (9) | 47 (6) | 0.305 |
| Chest or shoulder pain | 8 (9) | 10 (8) | 15 (13) | n/a | 15 (10) | 1 (1) | 7 (12) | n/a | 1 (6) | 4 (5) | 61 (8) | 0.08 |
| Other symptoms not listed above | 24 (28) | 44 (33) | 33 (29) | n/a | 64 (42) | 15 (17) | 7 (12) | n/a | 7 (41) | 20 (25) | 214 (29) | <0.001 |
| No symptoms | 5 (6) | 6 (5) | 6 (5) | n/a | 2 (1) | 31 (36) | 12 (20) | n/a | 0 (0) | 10 (12) | 72 (10) | <0.001 |
| Missing | 7 (8) | 7 (5) | 11 (10) | n/a | 16 (11) | 12 (14) | 6 (10) | n/a | 2 (12) | 10 (12) | 71 (10) | 0.392 |
| **Cancer-specificity of symptom presented** | | | | | | | | | | | | |
| Cancer-specific symptom | 62 (73) | 97 (73) | 79 (69) | n/a | 94 (62) | 36 (42) | 38 (62) | n/a | 10 (59) | 48 (59) | 464 (64) | <0.001 |
| Non-specific symptom | 11 (13) | 23 (17) | 19 (17) | n/a | 39 (26) | 7 (8) | 5 (8) | n/a | 5 (29) | 13 (16) | 122 (17) | |
| No symptoms/missing | 12 (14) | 13 (10) | 17 (15) | n/a | 18 (12) | 43 (50) | 18 (30) | n/a | 2 12) | 20 (25) | 143 (19) | |

All figures are n (%).
*Differences between jurisdictions (excluding Victoria and Norway) were tested by the Pearson's $\chi^2$ test, if nothing else stated.
†Differences between jurisdictions (excluding Victoria and Norway) were tested by the Fisher's exact test.
PCP, primary care physician.

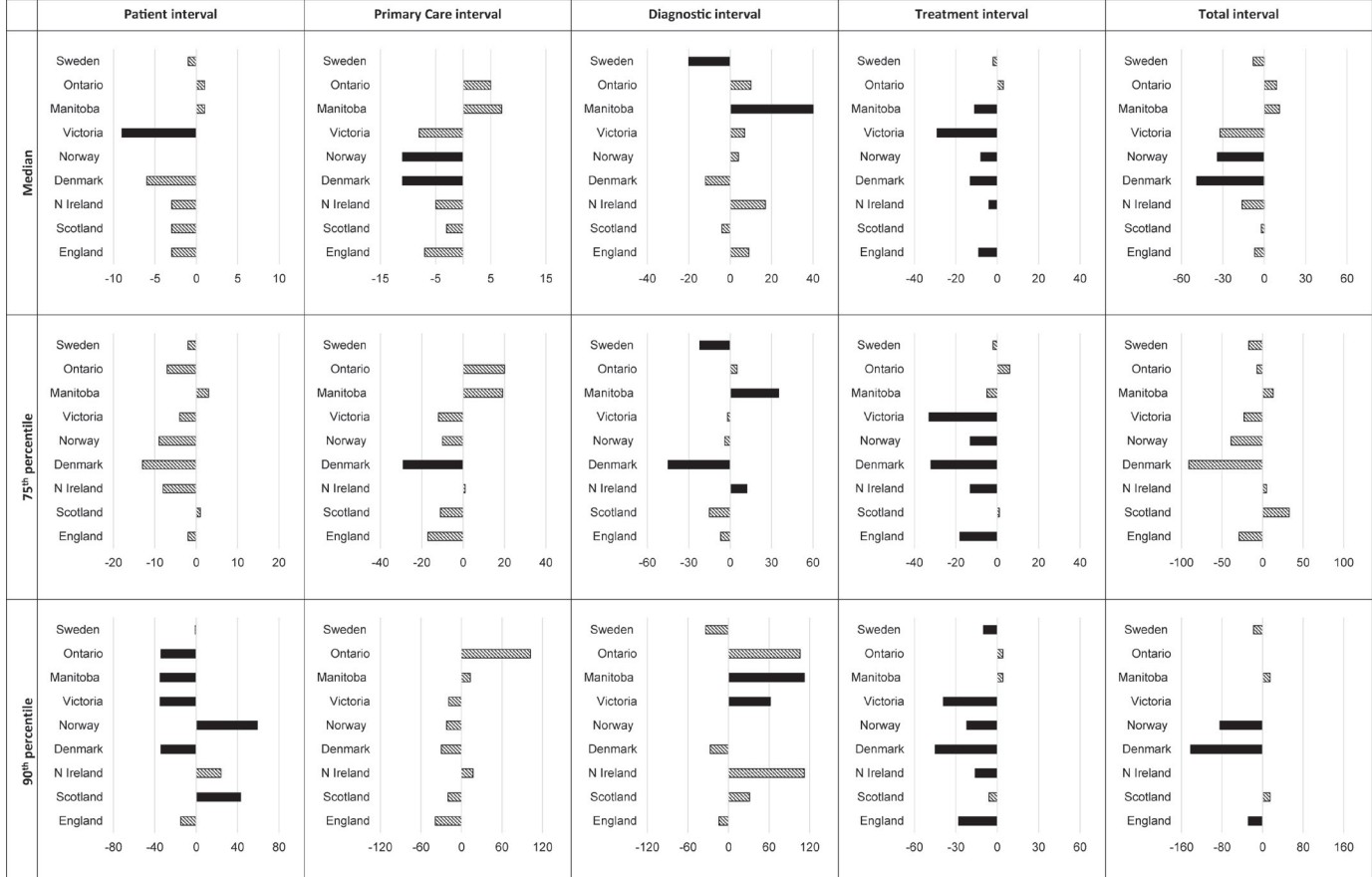

**Figure 2** Differences in 50th, 75th and 90th centiles of the intervals (days) between Wales as the reference and the other nine jurisdictions. The data are adjusted for differences in age, gender and comorbidity. The bars in black show significant differences in intervals.

survival differences.[38] The rate of respondents who attended A&E varied twofold across jurisdictions from 9%–10% in England, Scotland and Denmark to 18%–20% in Northern Ireland, Ontario and Manitoba. While rates for Scotland (10%) were similar to that reported in a prospective Scottish audit (11.5%), as were rates for Denmark (7% vs 6.3% when PCP not involved), rates for England (9%) were lower than those reported in population based audits (25%) reflecting non-response bias.[14 15] In Victoria (4%), restriction of the cohort to surgical patients is likely to have accounted for the very low rates.

Our reported median patient, primary care and diagnostic intervals are in keeping with those previously reported from the participating jurisdictions (table 6). Minor variations in interval estimates are likely due to differences in data source, sample size and cohort characteristics.[39] Longer intervals were reported from earlier cancer cohorts: median primary care interval for England of 52 days in 1998–2000 (our median 11),[13] median total interval for Denmark of 108 days in 2004–2005 (our median 67) and Norway of 118 in 2002–5 (our median 79).[17–19 24]

Across all jurisdictions, there was no significant difference in primary care intervals for the 10% of patients with longest interval. It is likely that these patients had

vague or non-specific symptoms and signs. Referral guidelines for suspected lung cancer do not always favour patients with early symptoms and often prioritise those with more advanced disease.[40] Access to better diagnostic tools such as low-dose CT chest in the primary care setting may favour this group of patients.[41] It would be useful in future projects to explore whether such access may have contributed to the improved 1-year lung cancer survival reported from Australia and Canada.[6]

Diagnostic intervals were significantly longer for Manitoba compared with other jurisdictions and twice that reported in an ongoing local PCP audit (personal communication). While one might suspect overestimation due to differences in the source of date of first presentation, between our study (in almost half, it was derived from patients) and local audit, this is less likely as the concordance coefficient between PCP and patient derived data at Manitoba was 0.94.

Observed median treatment intervals were below 6 weeks for nearly all jurisdictions. This was the only interval where there were significant differences between jurisdictions with Denmark, England, Norway and Northern Ireland all having shorter adjusted treatment intervals across all percentiles, with larger differences for the 75th and 90th percentile. These

**Table 5** Differences in adjusted intervals (days) between Wales and the other nine jurisdictions for patients with lung cancer

| Intervals | Percentiles | Wales Reference in days | Denmark Overall trend – shorter intervals | Sweden | England | Northern Ireland Similar with some intervals longer, some shorter | Scotland | Manitoba Overall trend – longer intervals | Ontario | Norway Difficult to interpret (see text for reasons) | Victoria |
|---|---|---|---|---|---|---|---|---|---|---|---|
| | Ranking by 5-year survival rates for lung cancers diagnosed in 1999-2007[5] | 10 | 6 | 1 | 9 | 7 | 8 | 3 | 2 | 5 | 4 |
| Patient interval | Number of patients | 181 | 233 | 172 | 213 | 179 | 169 | 133 | 205 | 55 | 141 |
| | 50th percentile (95% CI) | 21 | −6 (−13 to 0) | −1 (−9 to 8) | −3 (−10 to 5) | −3 (−13 to 6) | −3 (−10 to 4) | 1 (−8 to 10) | 1 (−11 to 14) | 0 (−8 to 8) | −9 (−16 to −2) |
| | 75th percentile (95% CI) | 61 | −13 (−38 to 13) | −2 (−24 to 21) | −2 (−42 to 37) | −8 (−55 to 39) | 1 (−25 to 27) | 3 (−23 to 30) | −7 (−54 to 39) | −9 (−60 to 42) | −4 (−46 to 38) |
| | 90th percentile (95% CI) | 216 | −34 (−55 to −13) | −1 (−25 to 23) | −15 (−42 to 12) | 24 (−21 to 70) | 43 (7 to 79) | −35 (−59 to −10) | −34 (−66 to −2) | 59 (21 to 96) | −35 (−49 to −21) |
| Primary care interval | Number of patients | 110 | 159 | N/A | 147 | 124 | 119 | 80 | 75 | 19 | 89 |
| | 50th percentile (95% CI) | 20 | −11 (−18 to −3) | | −7 (−17 to 3) | −5 (−15 to 4) | −3 (−14 to 8) | 7 (−8 to 21) | 5 (−9 to 19) | −11 (−18 to −4) | −8 (−17 to 1) |
| | 75th percentile (95% CI) | 43 | −29 (−47 to −12) | | −17 (−42 to 8) | 1 (−45 to 48) | −11 (−36 to 14) | 19 (−47 to 85) | 20 (−72 to 112) | −10 (−57 to 37) | −12 (−70 to 46) |
| | 90th percentile (95% CI) | 91 | −30 (−66 to 7) | N/A | −39 (−85 to 6) | 17 (−55 to 90) | −20 (−67 to 25) | 13 (−38 to 65) | 102 (−56 to 258) | −22 (−109 to 66) | −19 (−89 to 51) |
| Diagnostic interval | Number of patients | 176 | 229 | 165 | 212 | 170 | 173 | 138 | 212 | 52 | 160 |
| | 50th percentile (95% CI) | 45 | −12 (−25 to 1) | −20 (−35 to −5) | 9 (−3 to 21) | 17 (−5 to 39) | −4 (−16 to 8) | 40 (14 to 66) | 10 (−6 to 26) | 4 (−16 to 24) | 7 (−13 to 27) |
| | 75th percentile (95% CI) | 108 | −45 (−52 to −39) | −22 (−30 to −15) | −7 (−20 to 7) | 12 (2 to 22) | −15 (−32 to 1) | 35 (22 to 48) | 5 (−9 to 19) | −4 (15 to 8) | −2 (−8 to 4) |
| | 90th percentile (95% CI) | 162 | −27 (−153 to 99) | −34 (−206 to 138) | −14 (−100 to 72) | 112 (−165 to 389) | 31 (−81 to 143) | 112 (32 to 192) | 106 (−122 to 335) | 0 (−93 to 93) | 62 (10 to 114) |

Continued

**Table 5** Continued

| Intervals / Percentiles | Wales | Denmark | Sweden | England | Northern Ireland | Scotland | Manitoba | Ontario | Norway | Victoria |
|---|---|---|---|---|---|---|---|---|---|---|
| | Reference in days | Overall trend – shorter intervals | | | | Similar with some intervals longer, some shorter | Overall trend – longer intervals | | Difficult to interpret (see text for reasons) | |
| Ranking by 5-year survival rates for lung cancers diagnosed in 1999-2007[5] | 10 | 6 | 1 | 9 | 7 | 8 | 3 | 2 | 5 | 4 |
| **Treatment interval** — Number of patients | 192 | 279 | 190 | 238 | 200 | 187 | 182 | 263 | 87 | 199 |
| 50th percentile (95% CI) | 43 | −13 (−15 to −11) | −2 (−8 to 3) | −9 (−12 to −5) | −4 (−7 to −2) | 0 (−4 to 4) | −11 (−17 to −5) | 3 (−4 to 10) | −8 (−11 to −6) | −29 (−32 to −27) |
| 75th percentile (95% CI) | 64 | −32 (−36 to −28) | −2 (−8 to 4) | −18 (−23 to −13) | −13 (−18 to −7) | 1 (−7 to 9) | −5 (−16 to 6) | 6 (−2 to 14) | −13 (−19 to −8) | −33 (−41 to −25) |
| 90th percentile (95% CI) | 89 | −45 (−50 to −40) | −10 (−17 to −4) | −28 (−36 to −20) | −16 (−23 to −9) | −6 (−14 to 1) | 4 (−5 to 13) | 4 (−4 to 13) | −22 (−30 to −14) | −39 (−45 to −32) |
| **Total interval** — Number of patients | 147 | 192 | 147 | 176 | 153 | 143 | 117 | 178 | 52 | 113 |
| 50th percentile (95% CI) | 116 | −49 (−95 to −3) | −8 (−64 to 47) | −7 (−52 to 38) | −16 (−41 to 10) | −2 (−70 to 66) | 11 (−41 to 63) | 9 (−78 to 97) | −34 (−56 to −12) | −32 (−64 to 2) |
| 75th percentile (95% CI) | 204 | −91 (−270 to 87) | −17 (−40 to 7) | −29 (−175 to 118) | 5 (−191 to 201) | 33 (−144 to 211) | 13 (−77 to 103) | −7 (−331 to 317) | −39 (−107 to 29) | −23 (−61 to 14) |
| 90th percentile (95% CI) | 365 | −142 (−150 to −134) | −18 (−59 to 23) | −28 (−37 to −18) | 0 (−4 to 5) | 15 (−26 to 55) | 15 (−26 to 55) | 0 (−78 to 79) | −84 (−119 to −49) | 0 (−3 to 3) |

The majority of patients were diagnosed between 2013 and 2014. The differences were calculated for the 50th, 75th and 90th percentiles by setting age to its mean value and gender and comorbidity to their modes (ie, male gender and medium comorbidity). It is not possible to interpret differences observed for Norway (due to the small sample size) and Victoria (the cohort was limited to those who had undergone surgery).

**Table 6** Summary of literature on intervals in patients with lung cancer diagnosed since 2000 in the ICBP module 4 countries

| Study no | Ref | Study period | Jurisdiction | Design | Patients | No. of patients with lung cancer | Interval* (days) | | | | |
|---|---|---|---|---|---|---|---|---|---|---|---|
| | | | | | | | Patient | Primary care | Diagnostic | Treatment | Total interval |
| 1 | Walter et al[11] | 2011–2012 | England, UK | Prospective patient questionnaire survey – multihospital cohort. Dates of diagnosis based on medical note review. | All attending urgent and routine respiratory clinics across the five hospitals in England aged over 40 years with symptoms suspicious of lung cancer. | 153 | | | Interval from first symptom to diagnosis. Median 91 (IQR 49–184). | | |
| 2 | Lyratzopoulos et al | 2009–2010 | England, UK | Prospective national audit of cancer diagnosis using primary practice patient records and continuous sampling during audit period. | All patients aged >15 years who had first presented to a primary care practitioner and were subsequently diagnosed with 1 of 28 cancers. | 1128 | Median 11 (IQR 0–32). | Median 3 (IQR 14–39). | | | |
| 3 | Neal et al[12] | 2007–2008 | UK | Retrospective analysis of electonic health record data from General Practice Research Database – population cohort. | All newly diagnosed with 1 of 15 cancers. | 2851 | | Median 52 (IQR 7–243). | Median 112 (IQR 45–251). | | |
| 4 | Barrett and Hamilton[13] | 1998–2002 | Exeter, England, UK | Retrospective case-control review of PCP records – population cohort | All with lung cancer aged ≥40 years identified from the hospital cancer registry and computerised searches of all primary care practices. | 247 | | Interval from first symptom to diagnosis – Median 121 (IQR 53–261). | | | |
| 5 | Baughan et al[14] | 2005–2006, 2007–2008 | Scotland, UK | Retrospective audit involving PCP review of medical records of all newly diagnosed cancer patients they had seen – population cohort. | All newly diagnosed with lung cancer. | 981 | Median 9.5 (IQR 31). | | Median 11 (IQR 28). | | |
| 6 | Guldbrandt et al[15] | 2010 | Denmark | Retrospective PCP questionnaires survey of national registry-based population cohort. | All consecutive newly diagnosed patients with lung cancer. | 429–42 depending on interval. | | Median 7 (IQR 0–30). | Median 29 (IQR 12–69). | | |
| 7 | Torring et al[16] | 2004–2005 | Aarhus, Denmark | Prospective, population-based study using electronic health records and PCP survey of identified patients. | All newly diagnosed with lung cancer after attending primary care. | 262 | Median 28 (IQR 7–56). | Median 0 (IQR 0–9). | Median 52 (IQR 30–86). | | |
| 8 | Hansen et al[17] | 2004–2005 | Aarhus, Denmark | Retrospective PCP survey. | Cancer patients newly diagnosed during a 1-year period identified using administrative registry data. | 128–251 (depending on interval). | | | | Median 51 (IQR 27–76). | Median 108 (IQR 82–167). |

Continued

**Table 6** Continued

| Study no | Ref | Study period | Jurisdiction | Design | Patients | No. of patients with lung cancer | Patient | Primary care | Diagnostic | Treatment | Total interval |
|---|---|---|---|---|---|---|---|---|---|---|---|
| | | | | | | | **Interval* (days)** | | | | |
| 9 | Bjerager et al[18] | 2003 | Aarhus, Denmark | Retrospective PCP survey using structured telephone interviews enriched with administrative registry data – population-based cohort. | All patients with lung cancer identified through histological and cytological tests from county-based registers. | 84 | | Median 32.5 (IQR 12–68). | | | Median 118 (IQR 68–220). |
| 10 | Rolke et al (2006)[19] | 2002–2005 | Norway (South) | Retrospective questionnaire-based patient survey – hospital cohort. | All newly diagnosed with lung cancer. | 273–376 (depending on interval). | Median 19 (2–77). | | | | |
| 11 | Stokstad et al | 2011–2013 | Norway | Retrospective medical record audit – single hospital cohort. | All cases that started diagnostic work-up and were diagnosed with lung cancer at St. Olavs Hospital, Trondheim. | 449 | | | | 42 days (range: 2–296). | |
| 12 | Largey et al[20] | 2013 | Victoria, Australia | Retrospective medical record audit – three hospital cohorts. | Admitted with a new diagnosis of lung cancer over a 3-month period in three hospitals. | 78 | | | | Mean 30.4 (SD 45.3). | |
| 13 | Evans et al[21] | 2011–2014 | Victoria, Australia | Retrospective medical record audit – multihospital cohort. | All patients with lung cancer newly diagnosed in six public and two private hospitals. | 1417 | | | Median 15 (IQR 5–36). | Median 30 (IQR 6–84). | |
| 14 | Emery et al | 2012–2014 | Western rural Australia | Prospective cluster randomised trial of symptom awareness. | Patients with lung cancer newly diagnosed in the control arm of the trial. | 167 | Interval from first symptom to diagnosis. Median 34.5 (IQR 7–103.5). | | | | |
| 15 | Burmeister et al | 2000–2004 | Queensland, Australia | Retrospective analysis of radiation therapy waiting times. | All patients with lung cancer who received radiation therapy as initial treatment at a public hospital. | 1535 | | | | | Median 33† |
| 16 | Ellis and Vandermeer[22] | 2010 | Ontario, Canada | Retrospective patient survey using structured telephone interviews – single centre cohort. Appointment dates and diagnostic tests verified through family doctor or patient chart review. | All patients with lung cancer referred to a regional cancer centre. | 52 | | Median 21 | Median 27 (IQR 0–38). | | Median 138 (IQR 79–175). |
| 17 | Lo et al | 2005–2007 | Ontario, Canada | Retrospective medical record audit – multihospital cohort. | All with lung cancer seen on a newly implemented Time to Treat Program. | 144 | | | Median interval from suspicion of lung cancer to diagnosis: 37. | | |

*Intervals as defined in figure 1.
†Limited to patients receiving radiation treatment.
ICBP, International Cancer Benchmarking Partnership.

improvements may reflect implementation of waiting time targets in Denmark (35–38 days from first consultation depending on treatment modality) and the UK (31 days from decision to treat).[42 43] The shorter treatment intervals in Norway are in keeping with long-standing provision of standardised cancer care pathways and effective coordination between primary care and treatment centres. While a systematic review did not find evidence to support an association between intervals and lung cancer outcomes, increasing mortality with longer diagnostic intervals was noted in a more recent, high-quality study.[16] In 2000, O'Rourke and Edwards reported median intervals of 94 days (35–187) between the first hospital visit and starting treatment resulting in 21% of potentially curable patients becoming incurable.[44] Others have found metabolic evidence on PET/CT of pretreatment disease progression in 21% and TNM upstaging in 18% of small-cell lung cancer patients after a relatively short median interscan interval of 43 days.[45] Long intervals can also result in deterioration in performance status. More recently, there is concern that the need for genotyping may result in further increase in time to treatment.

The shorter total interval in Denmark likely reflects the significant reductions in cancer waiting times following a collaborative effort to set-up and implement a national centralised quality management system, the Danish Cancer Patient Pathways (CPPs). The latter includes PCP access to fast-track diagnostic work-up.[46] The findings are in keeping with higher relative survival and lower mortality in Denmark among symptomatic cancer patients diagnosed through primary care after the implementation of CPPs and with the accelerated increase in 5-year survival among Danish patients with lung cancer diagnosed in 2010–2014 when compared with patients from earlier time periods.[47 48] While there is some inherent lead-time bias, the findings highlight the importance and feasibility of a timely diagnosis of lung cancer.

## CONCLUSIONS

The study provides for the first time, comparable data, collected through consistent methods in all jurisdictions, allowing for detailed comparisons of key diagnostic intervals in lung cancer and routes to diagnosis. While all jurisdictions except Denmark had similar median adjusted total intervals, there were jurisdiction-specific significant differences in patient, diagnostic and treatment intervals, especially for the 10% of patients who waited the longest. The proportion of patients diagnosed following presentation to the PCP ranged from 35% to 75%. These data could help individual jurisdictions to better target their efforts to reduce time to treatment and ultimately improve patient experience and outcomes in lung cancer.

Intervals and pathways are ultimately of interest as they relate to prognosis. A further analysis that includes all four cancers (lung, ovary, colon and breast) surveyed in ICBP4 module and explores the impact of these intervals on stage and 1-year survival is underway.

**Author affiliations**
[1]Institute for Women's Health, University College London, London, UK
[2]Research Unit for General Practice, Aarhus University, Aarhus, Denmark
[3]Policy and Information, Cancer Research UK, London, UK
[4]Department of Prevention and Cancer Control, Cancer Care Ontario, Toronto, Ontario, Canada
[5]Centre for Behavioural Research in Cancer, Cancer Council Victoria, Melbourne, Victoria, Australia
[6]Department of General Practice, University of Melbourne, Melbourne, Victoria, Australia
[7]Centre for Population Health Sciences, Edinburgh University, Edinburgh, UK
[8]Scottish Cancer Registry, Information Services Division, NHS National Services Scotland, Edinburgh, UK
[9]The Royal Marsden, London, UK
[10]Institute for Cancer Research, Olso University Hospital, Oslo, Norway
[11]Department of Epidemiology and Cancer Registry, CancerCare Manitoba, Winnipeg, Manitoba, Canada
[12]Northern Ireland Cancer Registry, Queen's University Belfast, Belfast, UK
[13]Health Services Research Program, Ontario Institute for Cancer Research, Toronto, Ontario, Canada
[14]European Palliative Care Research Centre (PRC), Olso University Hospital, Oslo, Norway
[15]Institute of Clinical Medicine, University of Oslo, Oslo, Norway
[16]Department of Medical Epidemiology, Karolinska Institutet, Stockholm, Sweden
[17]Regional Oncologic Center, University Hospital, Uppsala, Sweden
[18]North Wales Centre for Primary Care Research, Bangor University, Wrexham, UK
[19]Department of Oncology, Lund University Hospital, Lund, Sweden
[20]Population Oncology, CancerCare Manitoba, Winnipeg, Manitoba, Canada
[21]Academic Unit of Primary Care, University of Leeds, Leeds, UK
[22]School of Psychology, Deakin University, Geelong, Victoria, Australia

**Acknowledgements** We are grateful to Catherine Foot, Martine Bomb and Brad Groves of Cancer Research UK for managing the programme. The International Cancer Benchmarking Partnership (ICBP) Module 4 Working Group (below) for their support and work keeping the study going. Christian Finley (Canada) and David Baldwin (England) for reviewing the manuscript. Stefan Bergström, Jan Willem Coebergh, Jon Emery, Monique E van Leerdam, Marie-Louise Essink-Bot and Una MacLeod (the ICBP M4 Academic Reference Group) for providing independent peer review of the study protocol and analysis plan development. We would like to thank all the patients, primary care physicians, cancer treatment specialists and registry staff of all jurisdictions who took part in this study and also the Danish Lung Cancer Registry for providing clinical information for the study.

**Collaborators** ICBP Module 4 Working Group: Alina Zalounina Falborg (Research Unit for General Practice, Department of Public Health, Aarhus University), Andriana Barisic (Department of Prevention and Cancer Control, Cancer Care Ontario), Anna Gavin (Northern Ireland Cancer Registry, Centre for Public Health, Queen's University Belfast), Anne Kari Knudsen (European Palliative Care Research Centre (PRC), Department of Oncology, Oslo University Hospital and Institute of Clinical Medicine), Breann Hawryluk (Department of Patient Navigation, Cancer Care Manitoba), Chantelle Anandan (Centre for Population Health Sciences, University of Edinburgh), Conan Donnelly (Centre for Public Health, Queen's University Belfast), David H Brewster (Scottish Cancer Registry, Information Services Division, NHS National Services Scotland, David Weller (Centre for Population Health Sciences, University of Edinburgh), Donna Turner (Population Oncology, Cancer Care Manitoba), Elizabeth Harland (Department of Epidemiology and Cancer Registry, CancerCare Manitoba), Eva Grunfeld (Knowledge Translation Research Network Health Services Research Program, Ontario Institute for Cancer Research; Professor and Vice Chair Research Department of Family and Community Medicine, University of Toronto), Evangelia Ourania Fourkala (Gynaecological Cancer Research Centre, Women's Cancer, Institute for Women's Health, University College London), Henry Jensen (Research Unit for General Practice, Department of Public Health, Aarhus University), Irene Reguilon (International Cancer Benchmarking Partnership, Cancer Research UK), Jackie Boylan (Centre for Public Health, Queen's University Belfast), Jacqueline Kelly (Northern Ireland Cancer Registry, Centre for Public Health, Queen's University Belfast), Jatinderpal Kalsi (Gynaecological Cancer Research Centre, Women's

Cancer, Institute for Women's Health, University College London), John Butler (The Royal Marsden) Kerry Moore, Centre for Public Health, Queen's University Belfast, Martin Malmberg (Department of Oncology, Lund University Hospital), Mats Lambe (Regional Cancer Center Uppsala and Department of Medical Epidemiology and Biostatics, Karolinska Institutet), Oliver Bucher (Department of Epidemiology and Cancer Registry, CancerCare Manitoba), Peter Vedsted, Research Unit for General Practice, Department of Public Health, Aarhus University, Rebecca-Jane Law (North Wales Centre for Primary Care Research, Bangor University), Rebecca Bergin (Centre for Behavioural Research in Cancer; Department of General Practice, University of Melbourne), Richard D Neal (North Wales Centre for Primary Care Research, Bangor University; Academic Unit of Primary Care, Leeds Institute of Health Sciences, University of Leeds), Samantha Harrison, Early Diagnosis and International Cancer Benchmarking Partnership, Policy and Information, Cancer Research UK, Sigrun Saur Almberg (Department of Cancer Research and Molecular Medicine, Faculty of Medicine, Norwegian University of Science and Technology (NTNU), Therese Kearney (Northern Ireland Cancer Registry, Centre for Public Health, Queen's University Belfast), Victoria Cairnduff (Northern Ireland Cancer Registry, Centre for Public Health, Queen's University Belfast), Victoria Hammersley (Centre for Population Health Sciences, University of Edinburgh), Victoria White (Centre for Behavioural Research in Cancer, Cancer Council Victoria Road; School of Psychology Deakin University), Usha Menon (Gynaecological Cancer Research Centre, Women's Cancer, Institute for Women's Health, University College London), Yulan Lin (European Palliative Care Research Centre (PRC), Department of Oncology, Oslo University Hospital and Institute of Clinical Medicine). ICBP Programme Board: Aileen Keel (Scottish Government, Edinburgh, Scotland); Anna Boltong (Cancer Council Victoria, Carlton, Australia); Anna Gavin (Northern Ireland Cancer Registry, Queen's University, Belfast, UK); David Currow (Cancer Institute New South Wales, Sydney, Australia); Gareth Davies (Wales Cancer Network, Cardiff, UK); Gunilla Gunnarsson (Swedish Association of Local Authorities and Regions, Stockholm, Sweden); Heather Bryant (Canadian Partnership Against Cancer, Toronto, Canada); Jane Hanson (Welsh Cancer National Specialist Advisory Group, Cardiff, UK); Kathryn Whitfield (Department of Health, Victoria Australia); Linda Rabeneck (Cancer Care Ontario, Toronto, Canada); Michael A Richards (Care Quality Commission, London, UK); Michael Sherar (Cancer Care Ontario, Toronto, Canada); Nicola Quin (Cancer Council Victoria, Carlton, Australia); Nicole Mittmann (Cancer Care Ontario, Toronto, Canada); Robert Thomas (Department of Health and Human Services, Victoria, Melbourne, Australia); Sara Hiom (Cancer Research UK); Sean Duffy (NHS England, London, UK); Chris Harrison (NHS England, London, UK); Søren Brostrøm (Danish Health and Medicines Authority, Copenhagen, Denmark); and Stein Kaasa (University Hospital of Trondheim, Trondheim, Norway). ICBP Academic Reference Group: Professor Jan Willem Coebergh, Professor of Cancer Surveillance, Department of Public Health, Erasmus Universiteit Rotterdam, Rotterdam, the Netherlands; Jon Emery, Professor of Primary Care Cancer Research, University of Melbourne and Clinical Professor of General Practice, University of Western Australia, Australia; Dr Stefan Bergström, senior consultant oncologist, Department of Oncology, Gävle, Sweden; Dr Monique E van Leerdam, Erasmus MC University Medical Centre, the Netherlands; Professor Marie-Louise Essink-Bot, Academic Medical Centre, Amsterdam University, the Netherlands; Professor Una MacLeod, Senior Lecturer in General Practice and Primary Care, Hull-York Medical School, UK.

**Contributors** UM, DW, PV, AZF and HJ planned the study design, data collection, carried out the analyses and wrote the draft manuscript. UM, PV, DW, HJ, AB, AKK, RJB, DHB, JK, VC, AG, EG, EH, ML, RJL, YL, MM, DT, RDN, VW, IR and SH were responsible for local data collection (alongside the Working Group), management and interpretation and have participated in writing and have approving the final manuscript version. JB, OB and OTB provided advice on the interpretation of results in their respective jurisdictions and comments or substantial edits on the manuscript and approving the final version.

**Funding** This work was supported by: CancerCare Manitoba; Cancer Care Ontario; Canadian Partnership Against Cancer; Cancer Council Victoria; Cancer Research Wales; Cancer Research UK; Danish Cancer Society; Danish Health and Medicines Authority; European Palliative Care Research Centre; Norwegian University of Science and Technology; Northern Ireland Guidelines Audit and Implementation Network; Macmillan Cancer Support; National Cancer Action Team; National Health Service (NHS) England; Medical Research Council (MR_UU_12023), Northern Ireland Cancer Registry, funded by the Public Health Agency Northern Ireland; Norwegian Directorate of Health; Research Centre for Cancer Diagnosis in Primary Care, Aarhus University, Denmark; Scottish Government; Swedish Association of Local Authorities and Regions; University College London and NIHR Biomedical Research Centre at University College London NHS Foundation Trust; University of Edinburgh; Victorian Department of Health and Human Services; and Welsh Government.

**Disclaimer** The funding bodies had no influence on the design of the study and collection, analysis and interpretation of data, in writing the manuscript or whether to publish the results.

**Competing interests** None declared.

**Patient consent for publication** Obtained.

**Ethics approval** For each local data collection, there were specific procedures and approvals that included anonymised data transfer to University College London and Aarhus University. Approvals were received from the following institutions: Cancer Council Victoria Human Research Ethics Committee (HREC 112); Health Research Ethics Board, University of Manitoba (HS15227 (H2012:105)); Research Resource Ethics Committee, CancerCare Manitoba (RRIC#28-2012); University of Toronto Research Ethics Board (27881); The Danish Data Protection Agency (2013-41-2030); Swedish Ethics Review Board, Uppsala (2013/306); Norway Regional committees for medical and health research ethics (2013/136/REK nord); England, Wales and Scotland, NRES Committee East Midlands – Derby 2, local R&D for each health board (11/EM/0420); and Northern Ireland ORECNI Ethical approval, local governance for each health Trust (11/EM/0420).

**Provenance and peer review** Not commissioned; externally peer reviewed.

**Data availability statement** The data that support the findings of this study are available from the named authors from each ICBP jurisdictions but restrictions apply to the availability of these data and so are not publicly available. Data are however available from the authors upon reasonable request and with permission of the ICBP Programme Board. Please contact the ICBP Programme Management team, based at Cancer Research UK, with any queries (icbp@cancer.org.uk).

**ORCID iDs**
Usha Menon http://orcid.org/0000-0003-3708-1732
Alina Zalounina Falborg http://orcid.org/0000-0002-1616-9455
Henry Jensen http://orcid.org/0000-0003-4040-7334
Irene Reguilon http://orcid.org/0000-0002-4246-6357
David H Brewster http://orcid.org/0000-0002-5346-5608

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
