## [Reviewer comments · BMJ Open]

ARTICLE DETAILS

TITLE (PROVISIONAL)	Time intervals and routes to diagnosis for lung cancer in ten jurisdictions: cross-sectional study findings from the International Cancer Benchmarking Partnership (ICBP)
AUTHORS	Menon, Usha; Vedsted, Peter; Zalounina Falborg, Alina; Jensen, Henry; Harrison, Samantha; Reguilon, Irene; Barisic, Andriana; Bergin, Rebecca; Brewster, David; Butler, John; Brustugun, Odd Terje; Bucher, Oliver; Cairnduff, Victoria; Gavin, Anna; Grunfeld, Eva; Harland, Elizabeth; Kalsi, Jatinderpal; Knudsen, Anne Kari; Lambe, Mats; Law, Rebecca-Jane; Lin, Yulan; Malmberg, Martin; Turner, Donna; Neal, Richard; White, Victoria; Weller, David

VERSION 1 – REVIEW

REVIEWER	Charles Helsper Julius Center, Utrecht University, University Medical Centre Utrecht, the Netherlands
REVIEW RETURNED	17-Sep-2018

GENERAL COMMENTS	Dear Authors, With great interest, I have read the manuscript “Time intervals and routes to diagnosis for lung cancer in ten jurisdictions: cross-sectional study findings from the International Cancer Benchmarking Partnership (ICBP)”, which was submitted for publication in BMJ open. It aims to report the first international comparison of routes to diagnosis and time intervals from symptom onset until treatment start for lung cancer patients. The manuscript describes an impressive undertaking, which addresses an important health care challenge. The presented information is an interesting and important addition to current knowledge, since international comparison of the duration of the durations of diagnostic pathways is important to unravel system-, disease- and patient-related factors that contribute to an unnecessarily prolonged patient journey. Even though I believe the presented information is valuable, I do have serious concerns concerning the effect of the methodology used on the interpretability and reliability of findings. The limitations causing these concerns are acknowledged and addressed by the authors. However, the solutions and explanations could not take away my concerns and ,in my view, deserve more attention. I will elaborate below. The major concern is that, in my view, the combination of non-
---

response bias and recall bias are likely to have a substantial influence on the findings in this study. Since the effect of this bias is hard to predict, it makes interpretation challenging.

Non response bias and selection is the biggest challenge for interpretation and usability of findings. If I understand correctly (1) only patients alive were contacted, (2) of these 70% were contacted, (3) one quarter of these (25.8%) completed a questionnaire, of whom (4) 81% was included in the analyses.

This raises questions, for instance those below. Defining the selection and the effects on all the (main) outcome measures is a challenge, which could be addressed in more detail. This seems necessary since this is vital for interpretation of the findings.

(1) Given the 1 year survival rate of under 50% for lung cancer, what is the effect of including only those alive on the findings? It could have different kinds of influence, but since the main hypothesis of this undertaking is that duration and pathways are related to prognosis (including burden and survival), by definition prognosis should be related to your main outcomes. This association is hard to predict reliably, but could go both ways for duration (alarm symptoms reduce duration vs delay causes bad prognosis) and pathways (more emergency for worse prognoses)?

(2) Why was only 30% of those alive not contacted (100%-70%)? Could this be a selected population?

(3) Who is this minority of 25% that responds? If I understand correctly, even within the "alive" population, these patients have a relatively good prognosis (low stage, high survival). And also fit enough to participate in research. Furthermore, table 1 shows that "Pack not forwarded by GP" occurs in 20% of cases. This is also very likely to be a selection. Why wouldn't the GP send the package to (burden) a selection of patients?

Minor remark on the subject; One solution presented in the strength and limitations on page 6 and on page 24 (Discussion, line 40), isn't comforting; "adjusting analyses for age, gender and comorbidity", is hardly a solution for non-response bias.

Recall bias seems to be of particular influence on the outcomes concerning duration. Asking patients to remember when a symptom, such as a cough, presented first, particularly after a tumultuous period after being diagnosed with lung cancer, does not seem to be a reliable way to obtain information. This is recognised but the solutions provided (e.g. <6 months after diagnosis and triangulation patient and GP memories), is only slightly comforting. Particularly since only 55% visited the GP with symptoms first.

The support for trustworthiness of this method, based on literature, could be more profound.

Also, triangulation is standardized but it actually makes it harder to fully understand the findings. Different sources were used for different patients, making it unclear to what extent which source provided the measurements for which patients. Can this be made more clear?

For a few (important) symptoms, it seems additionally challenging to determine the first occurrence. E.g. What is the “date of occurrence” of weight loss and persistent cough? This seems to be a change / process in time, not a moment. How was this defined and communicated to patients and PCPs?

Several remarks and solutions in the manuscript do not help to trust the reliability of findings and deserve explanation.

- Negative durations are set at 0 days and durations of over 365 days are truncated. This seems to imply that the authors do not really trust their own measurements. Furthermore, the solution to deal with these unexpected findings (truncation) seems suboptimal. Wouldn't omitting patients with unreliable findings be more appropriate than 'truncation to make sure the findings are within a reliable range'?

On the same subject; This method seems to be mentioned on Page 8, line 36. “All the measures were further validated using algorithms for outliers and implausible measures (e.g. negative time intervals).” I am not sure how using these algorithms ‘validates’ measures.

- On page 28, line 22, it states that “the diagnostic interval was twice that reported from ongoing local audit” . This is explained by “in 51% of the PCP responses presenting symptoms were missing or recorded as not present and date of first presentation was derived from patient as opposed to PCP”. Do I understand correctly that the missing data in this study may have led to a duration which is twice as long as the local audit? If so, this would indicate that missing data causes serious overestimation of duration in this study. If not, I do not understand this paragraph.

Some results are unclear. E.g. for the result in the abstract “All symptoms other than persistent cough were less frequently reported by the PCP when compared to patients.” It is unclear if these patients are all patients who actually visited the PCP (first). If not, the meaning of the difference could be explained in multiple ways with different meanings. If so, it seems unclear if this finding indicates e.g. that patients do not present all symptoms, if patients are subject to recall bias about their own symptoms in a different way, if patients are triggered differently by the questionnaire or something else. This is not addressed in the article and it seems hard to determine. This deserves (an attempt for) clarification.

The conclusions section on page 30 does not seem to reflect on the observation in a balanced way, since it doesn't follow the hierarchy and content of the primary and secondary outcomes. In my view, the conclusions deserves to be rewritten.

- The sentence “ Across countries there were discrepancies in symptoms, especially fatigue and weight loss reported by patients and their primary care physicians.” is unclear and seems out of place (too prominent). Is there a discrepancy in symptoms between countries or reporting between PCP?

- It states “The findings and quantifies achievements. “ Which “achievements” are quantified?

- “Thus allowing for more focused policy and practice initiatives” seems vague and not clearly supported by the findings.

- I don't really see how the last sentence “Meanwhile, our results draw attention to the success of secondary care initiatives in decreasing treatment intervals and underlines the need for more

	concerted efforts in primary care.” Is supported by your findings. Minor remarks  - The results section of the Abstract could (should?) provide more results on the primary and secondary outcome measures. For instance some information on actual interval lengths. - The conclusion of the abstract states “The data will allow jurisdictions to develop more focused lung cancer policy and clinical initiatives.” This seems to ambitious and could use a more modest tone. - In the strengths and limitations section it states “In Norway and Victoria, a small sample size and restriction of eligibility to only surgical patients, respectively, means some comparisons are made with caution – this mainly applies to the treatment interval and some patient characteristics.” > In my view, this selection may also substantially influence the diagnostic intervals. - In the references [“6-18”] to existing literature on the duration of lung cancer pathways (page 7 line 25), a reference to a relevant article is missing. It describes the duration of lung cancer intervals in the Netherlands: Time to diagnosis and treatment for cancer patients in the Netherlands: Room for improvement? Helsper CW, van Erp NF, Peeters PHM, de Wit NJ Dec 2017 In: European Journal of Cancer. 87, p. 113-121 9 p. - Typo on page 9, line 9. “..obvious discrepancy..” should be “..obvious discrepancies..”.
--	---

REVIEWER	Antoinette J. Wozniak MD Hillman Cancer Center University of Pittsburgh, United States
REVIEW RETURNED	12-Oct-2018

GENERAL COMMENTS	This manuscript describes the results of an international project to evaluate time intervals and routes to diagnosis of initial lung cancers. COMMENTS 1) Patients were collected from 10/2012-3/2015. Lung cancer screening was at least starting to be used toward the end of this period. Are there any data on whether any of these patients were screened. Might be interesting to note in which jurisdictions and at what time screening was implemented. Low dose CT is mentioned in the manuscript. Would like the authors opinion as to the possible impact of screening and whether future studies will attempt to evaluate its impact. 2) Smoking history was obtained from the patients on the questionnaire but not from the PCPs. It would seem that the knowledge of the smoking history may have influenced the PCP with regard to addressing symptoms earlier. Can the authors comment?
--

REVIEWER	Karolina Osowiecka Department of Public Health, University of Warmia and Mazury in Olsztyn; Department of Public Health, Medical University of Warsaw; Poland
REVIEW RETURNED	21-Oct-2018

GENERAL COMMENTS	Authors made a very interesting analysis indicating differences between jurisdictions in time intervals to lung cancer diagnosis and treatment especially looking for these differences for patients who wait the longest. The subject is worth exploring because there are lack of this kind of analysis. Authors showed the first so extensive, international, multicenter research. Some published results showed that the delay in waiting time influenced on survival prognosis of cancer patients. This is the reason why these research should be continuing to estimate the contribution differences in time interval from symptoms until treatment start for lung cancer patients and stage at diagnosis and survival. Authors estimated the different intervals of patient route. I have a suggestion to conduct another future analysis of defining factors which could influence the longer waiting time. In my opinion multivariate analysis of factors (in this manuscript there are a lot of examined) which could have impact on different intervals might be valuable and interesting for the readers.
---

REVIEWER	Ashanya Malalasekera Concord Repatriation General Hospital, Sydney, New South Wales, Australia
REVIEW RETURNED	28-Oct-2018

GENERAL COMMENTS	Thank you for conducting this very important, interesting and informative work. I congratulate all authors on this large, comprehensive effort, and echo all listed study strengths. Suggested revisions, comments and questions as follows: Abstract Page 4 Line 30 - The study design says "Patients, their primary care physicians (PCP) and cancer treatment specialists (CTS) were surveyed". However, results only convey details of patient response rate. Please include details of numbers of PCP and CTS contacted / responded and included in analysis. P4 L41 - if word count allows, please give the range of median total intervals or that of Wales. P7 L9 - Since study patients included Australians, worth mentioning 5 year survival in Australia also <20% (suggested ref: https://www.aihw.gov.au/reports/cancer/cancer-compendium-information-trends-by-cancer/report-contents/lung-cancer) P7 L12 - Perhaps fairer to revise this statement as "physician delays", given evidence that a GP will, on average, see only one case of lung cancer per year (Mansell 2011) and that delays occur in secondary healthcare as well (Neal 2015) P7 L 24-25: "While many national studies using different methodologies have reported on time intervals to treatment in lung cancer, as far as we are aware no international comparisons exist" - Are we sure about this statement? There are
---

data with international timeframe comparisons. Examples:
Olsson JK, Schultz EM, Gould MK. Timeliness of care in patients with lung cancer: a systematic review. *Thorax*. 2009;64(9):749-56
Malalasekera A, Nahm, S, Blinman PL, Kao SC, Dhillon H, Vardy J. How long is too long? A Scoping Review of Health System delays in lung cancer. *European Respiratory Reviews*. 2018; 29;27(149)
Jacobsen MM, Silverstein SC, Quinn M, Waterston LB, Thomas CA, Benneyan JC, et al. Timeliness of access to lung cancer diagnosis and treatment: A scoping literature review. *Lung Cancer-J Iaslc*. 2017;112:156-64.
Vinas F, Ben Hassen I, Jabot L, Monnet I, Chouaid C. Delays for diagnosis and treatment of lung cancers: a systematic review. *Clin Respir J*. 2016;10(3):267-71

Points of difference between this paper and cited refs above include it being an ICBP module substudy, weighted comparison using Wales as the reference jurisdiction, etc.

P7 L52 (and Supplementary File 4; Point11)- It is widely known that it is a significant challenge accurately identifying date of first presentation to healthcare. For example, in the chosen hierarchy of data sources for this date (Supp4, point 11), the PCP-yielded "date of date of first presentation to Primary Care and A&E" may reflect when the patient first presented to A&E with more overt/emergent symptoms of lung cancer (therefore resulting in expedited management and falsely shortened time interval) vs an earlier patient-yielded (potentially accurate) date of a more subtle presentation (resulting in longer time interval). Also, why are Primary Care and A&E classified together here? The date of patient presentation to either may not have been the same day in some cases. Also, wouldn't they represent different sectors of healthcare in some jurisdictions?

P7 L54 - Please use same terminology in the text / Tables as in Figure 1 eg "first presentation to health care" means "first contact with PCP"; "Primary Care interval" in Table 5 = "Delay in primary healthcare" in Figure 1. Also, the Diagnostic interval as per Aarhus statement needs to be defined (time from first clinical presentation to diagnosis). Finally, date of diagnosis may frequently precede the date of referral for treatment.

P7 L 18-21 - " Information on the types of treatment (surgery, chemotherapy, radiotherapy and other) were obtained from the patient survey." Wasn't this data also collected from specialists? (Suppl 3 Qn 5). While I read later that agreement between all sources was adequate (see later point), and although only 37% specialists responded (Table 1), any discrepancies in answers would warrant further study. Eg., in a responding patient population with less advanced disease, why did their doctors 'under'-report systemic symptoms such as fatigue and weight loss?

P8 L24 - I note the RECORD checklist used. What about the 35-item reporting checklist (1) used as the primary data source for the proposed REST guideline (2) currently under development for reporting studies on time to diagnosis?

(1) Launay E, Morfouace M, Deneux-Tharoux C, Gras le-Guen C, Ravaud P, Chalumeau M. Quality of reporting of studies evaluating time to diagnosis: a systematic review in paediatrics. *Arch Dis Child*. 2014;99(3):244-250.

(2) Reporting studies on time to diagnosis: proposal of a

	guideline by an international panel (REST). Enhancing the QUALity and Transparency Of health Research (EQUATOR) network: Centre for Statistics in Medicine, NDORMS, University of Oxford.; 2016.) P9 L6 (Figure 1) - For clarity, should include definition for 'Total Interval' given this is one of the primary outcomes. P12 Table 1 - Were any of the PCPs and/or CTS looking after >1 of the responding patients? P13 L4 - Replace "The characteristics" with "Patient characteristics" P23 L10 - Typo in table subheading - change "Overall" to "Overall" P24 L12 - "between all data sources": can we break this down further (or present the raw data) on agreement between patient vs PCP, and patient vs CTS? P22 L3 - Re: adjusted time intervals, how can we be totally sure of the primary data source for each date. For the sake of transparency / reproducibility, and given that time intervals were among the main study outcomes, can we see the raw data on how often rules were used in date interpretation? P25 L54 - 55: "suggesting that despite sampling issues, the pathway to diagnosis was comparable." Is this the pathway between two jurisdictions or between two jurisdictions and Wales? Why would they be comparable? P26 L6-10: Agree it is a limitation that only 26% respondents had Stage IV disease, which is not representative of the general population of patients newly diagnosed lung (75% with advanced disease). Is it possible to explore results for this subgroup alone? In Stage IV patients receiving both palliative radiation therapy and systemic therapy, which was considered by patients to be the 'first' cancer-specific treatment? What about palliative care referral and appointment? These are some considerations in the rapidly changing spectrum of management of lung cancer. P29 L34 and P30 L3 - I believe the Danish success story is partly due to the collaborative effort involved in setting up and maintaining a national, centralised quality management system. P29 L54 - Agree genotyping is a new element for consideration prior to management of advanced disease. But one may argue these are necessary requirements for optimal lung cancer care and evidence-based practice, and should not be considered a "delay". Which comes down to the question, what is the best way to define a 'delay' to lung cancer care? Would it be to benchmark against timeframe guidelines? Or patient-reported experience? This study uses Wales as the reference jurisdiction as it had the lowest lung cancer survival in ICBP Module 1 study, but the other side of the coin would be to compare against the jurisdiction with the shortest time interval, or best survival. Therein lies further issues that timeliness does not necessarily reflect better survival rates eg: the 'sicker quicker' hypothesis that management is expedited for symptomatic patients with advanced lung cancer / poorer survival rates.
--	---

REVIEWER	Mallikarjuna Rettiganti Senior Research Scientist, Eli Lilly and Company, Indianapolis (Since June 2018) Previously, Associate Professor of Pediatrics, University of Arkansas for Medical Sciences.
REVIEW RETURNED	06-Jan-2019

GENERAL COMMENTS	The authors compare time intervals and routes to diagnosis among ten jurisdictions using a cross-sectional study. The authors used appropriate design and methods to address study objectives. Limitations of the study are discussed clearly and extensively. However, there are a few minor suggestions that will help improve the manuscript. 1. In Table 4, the authors say they used a Chi-squared test for comparing proportions in each row. While reliable for large cell sizes, chi-squared test is not recommended for small cell sizes (when cell sizes are less than 5 usually). Some cell sizes are even 0. A Fisher's exact test is highly recommended. 2. Table 5. Showing direction and significance of differences between time intervals and 95% confidence intervals using different shades of gray is confusing (they are all shades of gray). I suggest using a figure (a forest plot or something similar where a vertical line at 0 would be a reference line for no difference), and decreases shown to the left, and increases shown to the right. These would be clearly interpretable. 3. Interestingly, the authors looked at median and other quantiles as opposed to mean time intervals. Why did the authors make this choice? That should be explained in the statistical methods section. Did the authors look at the distribution of these time intervals and determine they weren't normally distributed? The means would still be normally distributed given the sample size for most jurisdictions (central limit theorem). Also, a greater explanation regarding what comparing the 90th percentiles adds to the results, as opposed to comparing just the 50th percentiles, would be beneficial. Also what does it mean if the 90th percentile for one jurisdiction is higher (or lower) than the 90th percentile for another jurisdiction. On an average (50th percentile), time intervals are not significantly different between jurisdictions, but the 90th percentiles are different between jurisdiction? What does this mean. I believe, the results for the 50th percentiles and 90th percentiles could be tied together better in the discussion. 4. In the statistical methods, please add exactly what tests (functions and tests and related references) were used to do quantile regression and compare these percentiles. These would give more details and provide full disclosure. Overall, I think this paper throws light on the times taken from diagnosis to treatment for different areas.
---

VERSION 1 – AUTHOR RESPONSE

Reviewer 1

Comments	Response
The major concern is that, in my view, the combination of non-response bias and recall bias are likely to have a substantial influence on the findings in this study. Since the effect of this bias is hard to predict, it makes interpretation challenging. Non-response bias and selection is the biggest challenge for interpretation and usability of findings. If I understand correctly (1) only patients alive were contacted, (2) of these 70% were contacted, (3) one quarter of these (25.8%) completed a questionnaire, of whom (4) 81% was included in the analyses.	We agree with the reviewer's comments - the numbers outlined are correct and selection and non-response bias make interpretation challenging. This has been acknowledged in our discussion of study limitations and is now further emphasised However, it needs to be noted that there is no other way to gather much of the information on symptoms without conducting personal interviews with patients, and we sought large numbers in our study, making this impractical. Inherent in this kind of study is that only those who were alive could be contacted and ethics prevented us in certain countries from contacting patients whose primary practitioners deemed them too sick/anxious to participate in the survey. We have inserted in discussion paragraph three 'A key limitation, as with all questionnaire-based studies was both selection and non-response bias which varied across jurisdictions and has implications for interpretation and generalisation of findings.'
Given the 1 year survival rate of under 50% for lung cancer, what is the effect of including only those alive on the findings? It could have different kinds of influence, but since the main hypothesis of this undertaking is that duration and pathways are related to prognosis (including burden and survival), by definition prognosis should be related to your main outcomes. This association is hard to predict reliably, but could go both ways for duration (alarm symptoms reduce duration vs delay causes bad prognosis) and pathways (more emergency for worse prognoses)?	The author raises a key issue on impact of intervals and pathways on 1-year survival. There is a separate analysis that explores the association between health system intervals and 1 year-survival and this was stated in conclusion. Shorter intervals can, indeed, be associated with both better or worse survival We have expanded the statement in the conclusion 'Intervals and pathways are ultimately of interest as they relate to prognosis. A further analysis which includes all four cancers (lung, ovary, colon and breast) surveyed in ICBP4 module and explores the impact of these intervals on stage and 1-year survival is under consideration for publication.'
(2) Why was only 30% of those alive not contacted (100%-70%)? Could this be a selected population?	This was a pragmatic study and, inevitably, the regulations for contacting individuals varied between jurisdictions – we had to adjust our methods accordingly. Of those alive, ~30% were not contacted. This was not entirely random. Some of the reasons were stated in Table 1. This has now been further clarified in methods and elaborated in results. Methods 'In UK jurisdictions based on data protection laws, cancer registries could not directly contact patients.'

	Following a vital status check, cancer registries posted the patient questionnaire ... to the relevant primary care physician (PCP) who then forwarded the pre-addressed envelope to the patient after confirmation that the person was aware of the diagnosis and not deemed too sick/anxious to participate in the survey. (Wales, England, Scotland).’ Results ‘Of 4380 not contacted, 3367 (77%) were from England, Wales and Scotland. Major reasons reported by the PCP for not forwarding the survey included patients being terminally ill, not aware of cancer diagnosis at the time of request, having cognitive or visual impairment, language / communication difficulties, no longer at the address, not wishing to take part in research and a small number not having the index cancer. In addition, patents identified were not contacted in England as the target recruitment had been exceeded. For the non-UK jurisdictions, the main reasons for not contacting patients related to patients having died or no longer being at the address.’
(3) Who is this minority of 25% that responds? If I understand correctly, even within the “alive” population, these patients have a relatively good prognosis (low stage, high survival). And also fit enough to participate in research. Furthermore, table 1 shows that “Pack not forwarded by GP” occurs in 20% of cases. This is also very likely to be a selection. Why wouldn’t the GP send the package to (burden) a selection of patients?	It is well known that those who respond whether to a survey or to participate in a trial, are healthier than the remaining population – this is the healthy volunteer effort. It applies in all clinical studies/results involving informed consent. We do acknowledge this in the discussion in the strengths and weaknesses section. Nevertheless, this effect is likely to be similar in all jurisdictions, and we were principally interested in comparisons, rather than absolute values The reasons have been elaborated above
Minor remark on the subject; One solution presented in the strength and limitations on page 6 and on page 24 (Discussion, line 40), isn’t comforting; “adjusting analyses for age, gender and comorbidity”, is hardly a solution for non-response bias.	This was not meant as a solution for non-response and selection bias but as a separate limitation. The adjustment was to take into account differences in patient profile of the selected populations, not correct for selection bias While this was clear in Discussion ‘In comparing intervals, we adjusted for age, sex and comorbidity but were unable to adjust for ethnicity and education due to different classification systems’ in page 6 there was an error. This has been corrected.
Recall bias seems to be of particular influence on the outcomes concerning duration. Asking patients to remember	Recall bias was included in limitation and was minimised by the triangulation of different data sources and by patients completing the questionnaire within a

when a symptom, such as a cough, presented first, particularly after a tumultuous period after being diagnosed with lung cancer, does not seem to be a reliable way to obtain information. This is recognised but the solutions provided (e.g. <6 months after diagnosis and triangulation patient and GP memories), is only slightly comforting. Particularly since only 55% visited the GP with symptoms first. The support for trustworthiness of this method, based on literature, could be more profound. Also, triangulation is standardized but it actually makes it harder to fully understand the findings. Different sources were used for different patients, making it unclear to what extent which source provided the measurements for which patients. Can this be made more clear?	limited time window (median 5 months) after the cancer diagnosis. We agree that ascertaining time points in diagnostic journeys is challenging, particularly when symptoms are common, longstanding or non-specific. In developing our survey methods we drew on best available guidance to produce the most valid estimates of time points that we could: The Aarhus statement: improving design and reporting of studies on early cancer diagnosis. British journal of cancer. 2012 Mar;106(7):1262. We have added Supplementary Table 3 which shows, based on the standardization rules, which sources were used to define dates and how often a day in the date was imputed.
For a few (important) symptoms, it seems additionally challenging to determine the first occurrence. E.g. What is the “date of occurrence” of weight loss and persistent cough? This seems to be a change / process in time, not a moment. How was this defined and communicated to patients and PCPs?	The survey questionnaires are included in the supplementary appendix. Patients were asked to provide their best estimate of the date they noticed the first of the health concerns or symptoms. If they could not remember the exact date, they were asked to fill in the month and the year. The primary care physicians were asked to provide their best estimate of how long their patient had symptoms attributable to lung cancer before attending their practice.
Negative durations are set at 0 days and durations of over 365 days are truncated. This seems to imply that the authors do not really trust their own measurements. Furthermore, the solution to deal with these unexpected findings (truncation) seems suboptimal. Wouldn't omitting patients with unreliable findings be more appropriate than 'truncation to make sure the findings are within a reliable range'?	All survey-based studies of diagnostic journeys inevitably generate some responses which are difficult to interpret, but we consider that our steps taken to deal with ambiguous responses are robust: We have included Supplementary Table 4, which details the proportion of negative durations (which were set at 0 days) and durations of over 365 days (which were truncated) for each interval. We have repeated the analysis after excluding negative durations and durations of over 365 days and reported the results as a sensitivity analysis. Results – Sensitivity and validity analyses “Omitting time intervals which were negative or over 365 days (Supplementary Table 4) led to a change in direction of difference which was non-significant in long intervals (75th or 90th percentile) between Wales and jurisdictions in four cases: Norway and Victoria (patient interval), N Ireland (diagnostic interval), England (total interval). All other results were similar to the main results.’
On the same subject; This method seems	Thank you for drawing our attention to this sentence.

to be mentioned on Page 8, line 36. “All the measures were further validated using algorithms for outliers and implausible measures (e.g. negative time intervals).” I am not sure how using these algorithms ‘validates’ measures.	We understand that the sentence seems confusing, as the meaning did not stand out. We have now rephrased the sentence: Methods – Data handling “We applied rules for outliers and implausible measures (e.g. negative time intervals were recorded to zero-days and intervals longer than a year to 365 days).”
On page 28, line 22, it states that “the diagnostic interval was twice that reported from ongoing local audit” . This is explained by “in 51% of the PCP responses presenting symptoms were missing or recorded as not present and date of first presentation was derived from patient as opposed to PCP”. Do I understand correctly that the missing data in this study may have led to a duration which is twice as long as the local audit? If so, this would indicate that missing data causes serious overestimation of duration in this study. If not, I do not understand this paragraph.	We agree with the reviewer, that this paragraph is unclear and have rephrased it. There is, indeed, some evidence that PCPs and patients have different perceptions of key time points such as date of first presentation. However, while difference in data source might lead one to suspect overestimation in our study (compared to the audit), our concordance analyses of dates recorded by PCPs and patients suggest otherwise. We have included in Discussion ‘Diagnostic intervals were significantly longer for Manitoba compared to other jurisdictions and twice that reported in an ongoing local PCP audit (personal communication). While one might suspect overestimation due to differences in the source of date of first presentation, between our study (in just below half, it was derived from patients) and local audit (PCP based), this is less likely as the concordance co-efficient between PCP and patient derived data at Manitoba was 0.94.’
Some results are unclear. E.g. for the result in the abstract” All symptoms other than persistent cough were less frequently reported by the PCP when compared to patients.” It is unclear if these patients are all patients who actually visited the PCP (first). If not, the meaning of the difference could be explained in multiple ways with different meanings. If so, it seems unclear if this finding indicates e.g. that patients do not present all symptoms, if patients are subject to recall bias about their own symptoms in a different way, if patients are triggered differently by the questionnaire or something else. This is not addressed in the article and it seems hard to determine. This deserves (an attempt for) clarification.	We have clarified this further and included possible explanations. Results ‘When the analysis was restricted to the cohort where both patient and PCP had completed the survey, this difference persisted.’ Discussion ‘The median number of symptoms reported by patients was more than that reported by the PCP in all jurisdictions. This was especially so for fatigue and weight loss. A number of factors could have contributed to this for example, patients not listing all symptoms at presentation, patients having a different understanding/recall of their symptoms post diagnosis and PCPs only recording key symptoms such as cough.’
The conclusions section on page 30 does not seem to reflect on the observation in a balanced way, since it doesn’t follow the hierarchy and content of the primary and	The conclusion has been rewritten Conclusion The study provides for the first time, robust data,

secondary outcomes. In my view, the conclusions deserve to be rewritten.  - The sentence " Across countries there were discrepancies in symptoms, especially fatigue and weight loss reported by patients and their primary care physicians." is unclear and seems out of place (too prominent). Is there a discrepancy in symptoms between countries or reporting between PCP? - It states "The findings and quantifies achievements. " Which "achievements" are quantified? - "Thus allowing for more focused policy and practice initiatives" seems vague and not clearly supported by the findings. - I don't really see how the last sentence "Meanwhile, our results draw attention to the success of secondary care initiatives in decreasing treatment intervals and underlines the need for more concerted efforts in primary care." Is supported by your findings. 	collected through consistent methods in all jurisdictions, allowing for detailed comparisons of key diagnostic intervals in lung cancer and routes to diagnosis. While all jurisdictions except Denmark, had similar median adjusted total intervals, there were jurisdiction-specific significant differences in patient, diagnostic and treatment intervals, especially for the 10% of patients who waited the longest. The proportion of patients diagnosed following presentation to the PCP ranged from 35-75%. These data could help individual jurisdictions to better target their efforts to reduce time to treatment and ultimately improve patient experience and outcomes in lung cancer.' This sentence has been deleted.
Minor remarks  - The results section of the Abstract could (should?) provide more results on the primary and secondary outcome measures. For instance some information on actual interval lengths. -The conclusion of the abstract states "The data will allow jurisdictions to develop more focused lung cancer policy and clinical initiatives." This seems to ambitious and could use a more modest tone.  - In the strengths and limitations section it states "In Norway and Victoria, a small sample size and restriction of eligibility to only surgical patients, respectively, means some comparisons are made with caution – this mainly applies to the treatment interval and some patient characteristics." > In my view, this selection may also 	Abstract results have been revised as per recommendation 'With the exception of Denmark (-49 days), in all other jurisdictions the median adjusted total interval from symptom onset to treatment, for respondents diagnosed in 2012-15, was similar to that of Wales (116 days). Denmark had a shorter median adjusted primary care interval (-11 days) than Wales (20 days); Sweden had shorter (-20) and Manitoba longer (+40) median adjusted diagnostic intervals compared to Wales (45 days). Denmark (-13), Manitoba (-11), England (-9) and Northern Ireland (-4) had shorter median adjusted treatment intervals than Wales (43 days). The differences were greater for the 10% of patients who waited the longest.' Abstract conclusion has been revised to: 'The data could help jurisdictions develop more focused lung cancer policy and targeted clinical initiatives.' Revised 'The comparisons for Norway and Victoria, are limited by small sample size and inclusion of only surgical patients, respectively.'

substantially influence the diagnostic intervals. - In the references [“6-18”] to existing literature on the duration of lung cancer pathways (page 7 line 25), a reference to a relevant article is missing. It describes the duration of lung cancer intervals in the Netherlands: Time to diagnosis and treatment for cancer patients in the Netherlands: Room for improvement? Helsper CW, van Erp NF, Peeters PHM, de Wit NJ Dec 2017 In: European Journal of Cancer. 87, p. 113-121 9 p. - Typo on page 9, line 9. “..obvious discrepancy..” should be “..obvious discrepancies..”.	This has been added. Has been corrected.
--	--

Reviewer 2

1) Patients were collected from 10/2012-3/2015. Lung cancer screening was at least starting to be used toward the end of this period. Are there any data on whether any of these patients were screened. Might be interesting to note in which jurisdictions and at what time screening was implemented. Low dose CT is mentioned in the manuscript. Would like the authors opinion as to the possible impact of screening and whether future studies will attempt to evaluate its impact.	Thank you for your comments. National lung screening programmes were not in place in our jurisdictions during the period data was collected. There were however a few trials taking place in Denmark, the UK and Ontario. (https://www.ncbi.nlm.nih.gov/pmc/articles/PMC6037972/) All patients were asked to best describe the events that led to their diagnosis cancer. PCPs were also asked about the pathway of presentation. Any patient with a possible screen detected cancer was excluded from the analysis. There was only 1 screen detected patient in Ontario. The person was excluded from the analysis. We have clarified in abstract that these were symptomatic patients Abstract – Participants ‘Of 10,203 eligible symptomatic patients contacted, 2,631 (25.8%) responded and 2,143 (21.0%) were included in the analysis.’
2) Smoking history was obtained from the patients on the questionnaire but not from the PCPs. It would seem that the knowledge of the smoking history may have influenced the PCP with regard to addressing symptoms earlier. Can the authors comment?	We agree with the reviewer’s comment that knowledge of their patients smoking history may have influenced the PCP in this regard. However, we did not collect this information.

Reviewer 4

Thank you for conducting this very important, interesting and informative work. I congratulate all authors on this large, comprehensive effort, and echo all listed study strengths.	Thank you for your comments
Abstract Page 4 Line 30 - The study design says "Patients, their primary care physicians (PCP) and cancer treatment specialists (CTS) were surveyed". However, results only convey details of patient response rate. Please include details of numbers of PCP and CTS contacted / responded and included in analysis.	The percentages of PCP and CTS who responded are included in Table 1. We have now add this to abstract as well. Abstract – Participants ‘Data were also available from 1,211(56.6%) of their PCP and 643 (37.0%) of their CST.’
P4 L41 - if word count allows, please give the range of median total intervals or that of Wales. P7 L9 - Since study patients included Australians, worth mentioning 5 year survival in Australia also <20% (suggested ref: https://www.aihw.gov.au/reports/cancer/cancer-compendium-information-trends-by-cancer/report-contents/lung-cancer)	The medians for all intervals for all jurisdictions are included in Supplementary Web Table 2 in the Web Appendix. This has now been added
P7 L12 - Perhaps fairer to revise this statement as "physician delays", given evidence that a GP will, on average, see only one case of lung cancer per year (Mansell 2011) and that delays occur in secondary healthcare as well (Neal 2015)	Have removed primary care from statement. Introduction ‘Reasons for this are multi-faceted and include delays due to the atypical nature of some presenting symptoms, poor sensitivity of chest X-rays and physicians not acting quickly enough.’
P7 L 24-25: "While many national studies using different methodologies have reported on time intervals to treatment in lung cancer, as far as we are aware no international comparisons exist" - Are we sure about this statement? There are data with international timeframe comparisons. Examples: Olsson JK, Schultz EM, Gould MK. Timeliness of care in patients with lung cancer: a systematic review. Thorax. 2009;64(9):749-56 Malalasekera A, Nahm, S, Blinman PL, Kao SC, Dhillon H, VardyJ. How long is too long? A Scoping Review of Health System delays in lung cancer. European Respiratory Reviews. 2018; 29;27(149) Jacobsen MM, Silverstein SC, Quinn M, Waterston LB, Thomas CA, Benneyan JC, et al. Timeliness of access to lung cancer diagnosis and treatment: A scoping literature review. Lung Cancer-J Iaslc. 2017;112:156-64.	The reviewer has misunderstood. The named articles by the reviewer are reviews comparing intervals across studies – in contrast, ours is a single study comparing timeliness with the same methodology (survey instrument). Have now clarified in Introduction ‘Many national studies using different methodologies have reported on time intervals to treatment of lung cancer and there are reviews that have looked at international timeframe comparisons. [6-19] However, as far as we are aware there is no study that has undertaken international comparisons of timeliness across multiple countries using the same methodology.’ And will add to Discussion - Strengths and weaknesses ‘Our study helps address the shortcomings of

Vinas F, Ben Hassen I, Jabot L, Monnet I, Chouaid C. Delays for diagnosis and treatment of lung cancers: a systematic review. Clin Respir J. 2016;10(3):267-71 Points of difference between this paper and cited refs above include it being an ICBP module substudy, weighted comparison using Wales as the reference jurisdiction, etc.	current international comparisons across multiple national studies with significant variation in methodology including differences in definition of intervals.'
P7 L52 (and Supplementary File 4; Point11)- It is widely known that it is a significant challenge accurately identifying date of first presentation to healthcare. For example, in the chosen hierarchy of data sources for this date (Supp4, point 11), the PCP-yielded "date of date of first presentation to Primary Care and A&E" may reflect when the patient first presented to A&E with more overt/emergent symptoms of lung cancer (therefore resulting in expedited management and falsely shortened time interval) vs an earlier patient-yielded (potentially accurate) date of a more subtle presentation (resulting in longer time interval). Also, why are Primary Care and A&E classified together here? The date of patient presentation to either may not have been the same day in some cases. Also, wouldn't they represent different sectors of healthcare in some jurisdictions?	We agree with the reviewer and, as indicated, have drawn on best practice to maximise the validity of responses. Indeed, these issues are common to all studies in this area. It is because we acknowledge this that unlike in many studies, we have included the precise rules we used so that there is greater transparency. This makes it possible for individuals to better understand the data and how it has been analysed and interpreted. The classification includes the following routes - Symptoms prompting visit to PCP, Symptoms prompting emergency (A&E) department visit, Symptoms prompting visit to PCP and emergency (A&E) department While the number of patients who present to PCP but end up getting diagnosed through A&E are a minority (4% of patients with rates similar across jurisdictions) definitely they are a different subgroup from those diagnosed via PCP alone and the increased granularity of the data we present, will allow for greater ease of comparisons in the future reviews. For patients diagnosed via this route, the same standard rules were used – the date of patient presentation was the earliest date of presentation in the order of declining priority of below  a) date of first presentation to Primary Care from PCP data; b) date of first presentation to Primary Care and A&E from PCP data; c) date of first presentation to Primary Care from patient data. The earlier patient-yielded (potentially accurate) date of a subtler presentation (resulting in longer time interval) could have been reported by the PCP but there may be a few cases when it was missed. In all jurisdictions, we defined PCP and used a standard definition of A&E / emergency/ casualty

	that was agreed with the local team.
P7 L54 - Please use same terminology in the text / Tables as in Figure 1 eg "first presentation to health care" means "first contact with PCP"; "Primary Care interval" in Table 5 = "Delay in primary healthcare" in Figure 1. Also, the Diagnostic interval as per Aarhus statement needs to be defined (time from first clinical presentation to diagnosis). Finally, date of diagnosis may frequently precede the date of referral for treatment.	Definitions are included in Figure 1. And terminology is consistent across Figure and text.
P7 L 18-21 - " Information on the types of treatment (surgery, chemotherapy, radiotherapy and other) were obtained from the patient survey." Wasn't this data also collected from specialists? (Suppl 3 Qn 5). While I read later that agreement between all sources was adequate (see later point), and although only 37% specialists responded (Table 1), any discrepancies in answers would warrant further study. Eg., in a responding patient population with less advanced disease, why did their doctors 'under'-report systemic symptoms such as fatigue and weight loss?	Given the agreement on treatment, we consider that use of patient data is warranted. We agree where there are discrepancies in areas such as symptoms; these warrant further research and we have included this in discussion. Discussion: 'Further research on 'under reporting of systemic symptoms such as fatigue and weight loss is warranted.'
P8 L24 - I note the RECORD checklist used. What about the 35-item reporting checklist (1) used as the primary data source for the proposed REST guideline (2) currently under development for reporting studies on time to diagnosis? (1) Launay E, Morfouace M, Deneux-Tharoux C, Gras le-Guen C, Ravaud P, Chalumeau M. Quality of reporting of studies evaluating time to diagnosis: a systematic review in paediatrics. Arch Dis Child. 2014;99(3):244-250. (2) Reporting studies on time to diagnosis: proposal of a guideline by an international panel (REST). Enhancing the Quality and Transparency Of Health Research (EQUATOR) network: Centre for Statistics in Medicine, NDORMS, University of Oxford.; 2016.)	We have not used the REST checklist. As the reviewer indicates it is in development. In fact we double checked at www.equator-network.org, and the REST guideline is noted to be under validation (http://www.equator-network.org/wp-content/uploads/2009/02/Reporting-studies-on-time-to-diagnosis-summary.pdf). Furthermore, the REST guideline has, after the publication in 2016, only been cited twice (one is a self-citation). We could find no other reference to it on a google search.
P9 L6 (Figure 1) - For clarity, should include definition for 'Total Interval' given this is one of the primary outcomes. P12 Table 1 - Were any of the PCPs and/or CTS looking after >1 of the responding	Figure 1 now includes total interval It is possible that they were. Only anonymised data was shared centrally so it is not possible to provide exact numbers.

patients? P13 L4 - Replace "The characteristics" with "Patient characteristics" P23 L10 - Typo in table subheading - change "Overall" to "Overall" P24 L12 - "between all data sources": can we break this down further (or present the raw data) on agreement between patient vs PCP, and patient vs CTS?	Have been addressed Corrected We have added the results for agreement between patient vs PCP, and patient vs CTS Results - sensitivity analyses 'Agreement between patient versus PCP for dates of first presentation to primary care (CCC=0.91) and diagnosis (CCC=0.93) was also adequate as was agreement between patient versus CTS for dates of diagnosis (CCC=0.94) and treatment (CCC=0.94).'
P22 L3 - Re: adjusted time intervals, how can we be totally sure of the primary data source for each date. For the sake of transparency / reproducibility, and given that time intervals were among the main study outcomes, can we see the raw data on how often rules were used in date interpretation? P25 L54 - 55: "suggesting that despite sampling issues, the pathway to diagnosis was comparable." Is this the pathway between two jurisdictions or between two jurisdictions and Wales? Why would they be comparable? P26 L6-10: Agree it is a limitation that only 26% respondents had Stage IV disease, which is not representative of the general population of patients newly diagnosed lung (75% with advanced disease). Is it possible to explore results for this subgroup alone? In Stage IV patients receiving both palliative radiation therapy and systemic therapy, which was considered by patients to be the 'first' cancer-specific treatment? What about palliative care referral and appointment? These are some considerations in the rapidly changing spectrum of management of lung cancer.	We have added a Supplementary Web Table 3, which shows, based on the standardization rules, which sources were used to define dates and how often a day in the date was imputed. Also, we have shown the amount of negative durations (which were set at 0 days) and durations of over 365 days (which were truncated) (Supplementary Web Table 4). The sentence has been rephrased. 'In Norway and Victoria, a small sample size and restriction of eligibility to only surgical patients, respectively, made comparison difficult. Nonetheless, significant differences in these two jurisdictions compared to Wales were largely limited to the treatment interval alone.' The study group are currently running a separate analysis of intervals and stage. We agree that sub analyses of the stage IV population, as outlined by the reviewer would be helpful. However, they are subject to loss of power, making inferences difficult.
P29 L34 and P30 L3 - I believe the Danish success story is partly due to the collaborative effort involved in setting up and maintaining a national, centralised quality management system.	We have added this to the discussion: 'The shorter total interval in Denmark likely reflects the significant reductions in cancer waiting times following a collaborative effort to set-up and implement a national centralised quality management system, the Danish Cancer Patient Pathways (CPPs).'
P29 L54 - Agree genotyping is a new element	We agree on this with the reviewer. We have

for consideration prior to management of advanced disease. But one may argue these are necessary requirements for optimal lung cancer care and evidence-based practice and should not be considered a "delay". Which comes down to the question, what is the best way to define a 'delay' to lung cancer care? Would it be to benchmark against timeframe guidelines? Or patient-reported experience? This study uses Wales as the reference jurisdiction as it had the lowest lung cancer survival in ICBP Module 1 study, but the other side of the coin would be to compare against the jurisdiction with the shortest time interval, or best survival. Therein lies further issues that timeliness does not necessarily reflect better survival rates eg: the 'sicker quicker' hypothesis that management is expedited for symptomatic patients with advanced lung cancer / poorer survival rates.	therefore in our own study used the word 'interval' instead of 'delay' where possible. Indeed, the term 'delay' carries negative and unhelpful connotations about diagnostic and treatment practices We have changed the wording in Discussion 'More recently, there is concern that the need for genotyping may result in further increase in time to treatment.' A separate analysis of intervals and survival has been submitted for publication which highlights the 'sicker quicker' hypothesis that management is expedited for symptomatic patients with advanced lung cancer / poorer survival rates.
---	--

Reviewer 5

The authors compare time intervals and routes to diagnosis among ten jurisdictions using a cross-sectional study. The authors used appropriate design and methods to address study objectives. Limitations of the study are discussed clearly and extensively. However, there are a few minor suggestions that will help improve the manuscript.	
1. In Table 4, the authors say they used a Chi-squared test for comparing proportions in each row. While reliable for large cell sizes, chi-squared test is not recommended for small cell sizes (when cell sizes are less than 5 usually). Some cell sizes are even 0. A Fisher's exact test is highly recommended. 2. Table 5. Showing direction and significance of differences between time intervals and 95% confidence intervals using different shades of gray is confusing (they are all shades of gray). I suggest using a figure (a forest plot or something similar where a vertical line at 0 would be a reference line for no difference), and decreases shown to the left, and increases shown to the right. These would be clearly interpretable.	We have added Fisher's exact test to Table 4 and the statistical methods section. Figure 2 as described by the reviewer is included Table 5 gives precise confidence intervals and are likely to appeal to a different section of readers. Our preference is to use both but ensure they are side by side.
3. Interestingly, the authors looked at median and other quantiles as opposed to mean time intervals. Why did the authors make this choice? That should be explained in the statistical methods section. Did the authors look at the distribution of these time intervals and determine they weren't normally distributed? The means would still be	An explanation, why the results of the study were reported in terms of the 50th/75th/90th interval percentiles, has been added. Methods – statistical analysis 'The differences in intervals between the jurisdictions were estimated using quantile regression, as this method allows for a comparison across the whole

normally distributed given the sample size for most jurisdictions (central limit theorem). Also, a greater explanation regarding what comparing the 90th percentiles adds to the results, as opposed to comparing just the 50th percentiles, would be beneficial. Also what does it mean if the 90th percentile for one jurisdiction is higher (or lower) than the 90th percentile for another jurisdiction. On an average (50th percentile), time intervals are not significantly different between jurisdictions, but the 90th percentiles are different between jurisdiction? What does this mean. I believe, the results for the 50th percentiles and 90th percentiles could be tied together better in the discussion.	distribution of length of the interval.[25] As we were interested in a measure of central tendency of length of the interval and in long- and very long intervals, the focus of the study was on the 50th(median), 75th and 90th interval percentiles. Wales was chosen as the reference jurisdiction as it had the lowest lung cancer survival in ICBP Module 1 analysis.' This has been addressed Abstract 'Denmark had a shorter median adjusted primary care interval (-11 days) than Wales (20 days); Sweden had shorter (-20) and Manitoba longer (+40) median adjusted diagnostic intervals compared to Wales (45 days). Denmark (-13), Manitoba (-11), England (-9) and Northern Ireland (-4) had shorter median adjusted treatment intervals than Wales (43 days). The differences were greater for the 10% of patients who waited the longest. 'There are differences between jurisdictions in interval lengths to treatment, which are magnified in lung cancer patients who wait the longest.'
4. In the statistical methods, please add exactly what tests (functions and tests and related references) were used to do quantile regression and compare these percentiles. These would give more details and provide full disclosure.	More details on quantile regression have been added. Methods – statistical analysis 'Since the length of the interval in days is a continuous measure which has been rounded, we applied the quantile regression analysis on the smoothed quantiles; the method based on the smoothed quantiles is recommended for analyses of discrete (count) data [26]. In STATA this method is implemented in the 'qcount' procedure. [27] Parameters were calculated with 1000 jittered samples. For all interval analyses, the differences in intervals were calculated as marginal effects after quantile regression by setting the continuous covariate (age) to their mean values and the categorical covariates (sex and comorbidity) to their modes.'

VERSION 2 – REVIEW

REVIEWER	Charles Helsper Julius center for health sciences and primary care, UMC Utrecht / Utrecht University. Utrecht, The Netherlands
REVIEW RETURNED	29-May-2019

GENERAL COMMENTS	Second review of : “Time intervals and routes to diagnosis for lung cancer in ten jurisdictions: cross-sectional study findings from the International Cancer Benchmarking Partnership (ICBP)” Dear Authors, Thank you for the revisions on your manuscript “Time intervals and routes to diagnosis for lung cancer in ten jurisdictions: cross-sectional study findings from the International Cancer Benchmarking Partnership (ICBP)”. I believe this paper carries important messages, which could have considerable implications, making interpretability of paramount importance. This is why my previous review of this report of an impressive undertaking was quit stern. In my view, the additions, clarifications and corrections made by the authors (mainly addressing limitations) have substantially improved interpretability. With the changes made, combined with the availability of the review alongside the paper, I believe the consequences of the studies limitations for interpretation have been sufficiently addressed. Only two minor comments:  - Changing the word “comparable” to “robust” in the first line of the conclusions sections seems a bit too bold. I believe comparable is better justifiable. - The following references could be useful. I would like leave it to the consideration of the authors if they consider referring to these articles of additional value.  • (1) How long is too long? A scoping review of health system delays in lung cancer. Ashanya Malalasekera, Sharon Nahm, Prunella L. Blinman, Steven C. Kao, Haryana M. Dhillon, Janette L. Vardy. European Respiratory Review 2018 27: 180045; DOI: 10.1183/16000617.0045-2018 • (2) The Aarhus statement on cancer diagnostic research: turning recommendations into new survey instruments. Domenica Coxon, Christine Campbell, Fiona M. Walter, Suzanne E. Scott, Richard D. Neal, Peter Vedsted, Jon Emery, Greg Rubin, William Hamilton and David Weller BMC Health Services Research201818:677 https://doi.org/10.1186/s12913-018-3476-0
--

REVIEWER	Ashanya Malalasekera Concord Cancer Centre, Concord Repatriation General Hospital, NSW, Australia
REVIEW RETURNED	20-May-2019

GENERAL COMMENTS	Thank you for your time, conscientious revisions and efforts. My main comments are re: the treatment start (detail and date), from
--

	Supplementary Appendix C3 ("Data sources used to define dates and percentage of imputed dates") and Appendix B "Rules for missing, incomplete, multiple response and out of range data." It is surprising that the majority (55%) of dates for treatment start (an endpoint to multiple key time intervals) were from patient source, and only 45% from registry/CST medical records, which (I expect) would be electronically captured and thus less subject to error. Could authors comment? On a related point, What is "Other" as a form of treatment on Pg 33 (Rule 14: Date of treatment start from patient data is defined as the earliest of the treatment dates for Surgery, Chemo, Radio and Other;" - could it include palliative care? I understand the study group are conducting a separate analysis of intervals and stage, but even patients with locally advanced lung cancer, may also be referred to Palliative care for assistance with symptom control.
--	---

VERSION 2 – AUTHOR RESPONSE

Reviewer 4; Name: Ashanya Malalasekera; Institution and Country: Concord Cancer Centre, Concord Repatriation General Hospital, NSW, Australia

My main comments are re: the treatment start (detail and date), from Supplementary Appendix C3 ("Data sources used to define dates and percentage of imputed dates") and Appendix B "Rules for missing, incomplete, multiple response and out of range data."

It is surprising that the majority (55%) of dates for treatment start (an endpoint to multiple key time intervals) were from patient source, and only 45% from registry/CST medical records, which (I expect) would be electronically captured and thus less subject to error. Could authors comment?

Many thanks for your comments.

We agree that registry/CST medical records are most probably less subject to error (therefore this data source was chosen as the first priority in the definition of date of treatment). However, registry/CST medical records on date of treatment were not available for 55% patients, and an alternative data source (patient survey) was used instead. We have added this clarification to Supplementary Table 3.

Supplementary Table 3: Data sources used to define dates and percentage of imputed dates

Type of date	Data sources used to define a date* (%)				Cases with imputed day in a date** (%)
	Patient	PCP	CST	Registry	
First noticing symptoms	100	0	0	0	66
First presentation to health care	49	51	0	0	30
First referral to secondary care	0	100	0	0	1
Diagnosis	5	6	8	81	1
Start of curative or palliative treatment	55***	0	32	13	11

* based on rules 10-14, supplementary file Appendix B

** based on rule 15, supplementary file Appendix B

*** Registry/CST medical records on date of treatment were not available for 55% patients, therefore an alternative data source (patient survey) was used instead.

On a related point, What is "Other" as a form of treatment on Pg 33 (Rule 14: Date of treatment start from patient data is defined as the earliest of the treatment dates for Surgery, Chemo, Radio and Other;" - could it include palliative care? I understand the study group are conducting a separate analysis of intervals and stage, but even patients with locally advanced lung cancer, may also be referred to Palliative care for assistance with symptom control.

The 'other' response from patients included palliative care as rightly inferred by the referee, participation in a clinical trial, targeted agents like erlotinib and procedures like pleural tap. This has been added to Rule 14 on page 33 of Appendix.

Reviewer: 1

Reviewer Name: Charles Helsper; Institution and Country: Julius center for health sciences and primary care, UMC Utrecht / Utrecht University. Utrecht, The Netherlands

Please leave your comments for the authors below Second review of : "Time intervals and routes to diagnosis for lung cancer in ten jurisdictions: cross-sectional study findings from the International Cancer Benchmarking Partnership (ICBP)"

Dear Authors,

Thank you for the revisions on your manuscript "Time intervals and routes to diagnosis for lung cancer in ten jurisdictions: cross-sectional study findings from the International Cancer Benchmarking Partnership (ICBP)".

I believe this paper carries important messages, which could have considerable implications, making interpretability of paramount importance. This is why my previous review of this report of an impressive undertaking was quit stern.

In my view, the additions, clarifications and corrections made by the authors (mainly addressing limitations) have substantially improved interpretability.

With the changes made, combined with the availability of the review alongside the paper, I believe the consequences of the studies limitations for interpretation have been sufficiently addressed.

Many thanks for your comments

Only two minor comments:

-Changing the word "comparable" to "robust" in the first line of the conclusions sections seems a bit too bold. I believe comparable is better justifiable.

Agree, now updated.

-The following references could be useful. I would like leave it to the consideration of the authors if they consider referring to these articles of additional value.

•(1) How long is too long? A scoping review of health system delays in lung cancer. Ashanya Malalasekera, Sharon Nahm, Prunella L. Blinman, Steven C. Kao, Haryana M. Dhillon, Janette L. Vardy. *European Respiratory Review* 2018 27: 180045; DOI: 10.1183/16000617.0045-2018

We have not included this as we have already included systematic reviews on the subject.

•(2) The Aarhus statement on cancer diagnostic research: turning recommendations into new survey instruments. Domenica Coxon, Christine Campbell, Fiona M. Walter, Suzanne E. Scott, Richard D.

Neal, Peter Vedsted, Jon Emery, Greg Rubin, William Hamilton and David Weller BMC Health Services Research 2018;18:677

We have added this reference to the limitations section.